# Exploring the Frontiers of Softmax: Provable Optimization, Applications in Diffusion Model, and Beyond

## Abstract

The softmax activation function plays a crucial role in the success of large language models (LLMs), particularly in the self-attention mechanism of the widely adopted Transformer architecture. However, the underlying learning dynamics that contribute to the effectiveness of softmax remain largely unexplored. As a step towards better understanding, this paper provides a theoretical study of the optimization and generalization properties of two-layer softmax neural networks, providing theoretical insights into their superior performance as other activation functions, such as ReLU and exponential. Leveraging the Neural Tangent Kernel (NTK) framework, our analysis reveals that the normalization effect of the softmax function leads to a good perturbation property of the induced NTK matrix, resulting in a good convex region of the loss landscape. Consequently, softmax neural networks can learn the target function in the over-parametrization regime. To demonstrate the broad applicability of our theoretical findings, we apply them to the task of learning score estimation functions in diffusion models, a promising approach for generative modeling. Our analysis shows that gradient-based algorithms can learn the score function with a provable accuracy. Our work provides a deeper understanding of the effectiveness of softmax neural networks and their potential in various domains, paving the way for further advancements in natural language processing and beyond.

## 1 Introduction

Large Language Models (LLMs) like GPT4 (Achiam et al., 2023) from OpenAI and Claude 3 (Anthropic, 2024) from Anthropic have widely and profoundly changed the world. Some researchers believe they split human history into two parts: the Pre-LLM Era and the LLM Era. The LLMs have been widely used in human activities, such as education (Kasneci et al., 2023), law (Sun, 2023), finance (Li et al., 2023c), bio-informatics (Thirunavukarasu et al., 2023), coding (Hou et al., 2024), and even top AI conference reviews such as ICML, ICLR, NeurIPS, and AISTATS (Liang et al., 2024a). To make LLMs successful, one of the cores of LLMs is the Transformer model architecture (Vaswani et al., 2017), which has many advantages, including faster-parallelized inference rather than sequential inference like RNN (Hochreiter & Schmidhuber, 1997); being easy to scale up the model capacity to support the scaling laws in neural language models (Kaplan et al., 2020), i.e. since the input and output dimension of each Transformer blocks is the same, we can stack an arbitrary number of layers as we want. The kernel design of the Transformer block is self-attention layers, where each block has many attention heads and each head has its three important private parameter matrices for key, query, and value operation. Many papers believe that the self-attention operation is the critical reason for emergent ability (Wei et al., 2022), including in-context learning (Olsson et al., 2022; Reddy, 2024) and compositional ability to solve complex task (Dziri et al., 2024; Lu et al., 2024). The Transformer is so successful and has been widely certified that this architecture can be adopted in many other modalities such as tabular data, image/video generation, e.g., the video diffusion model SORA (OpenAI, 2024) using Transformer (Peebles & Xie, 2023) as its backbone.

When we delve into the self-attention mechanism, we find the softmax function plays a crucial role (Vaswani et al., 2017). It enables the model to focus on the most related information among

Table 1: Comparing hidden neuron number $m$ in two-layer neural networks and training steps $\widehat{T}$ are required under different activation functions to guarantee that, for any $\epsilon > 0$, with probability at least 0.99, the training loss is smaller or equal to $\epsilon$. Here, $n$ is the number of training samples, and $\lambda$ is the smallest eigenvalue for the matrix of the neural tangent kernel, where $n > 1$ and $\lambda < 1$. We can see that the two-layer NN with softmax activation function requires almost the same number of neurons and training steps to converge as that with ReLU or exponential activation functions. More details: Theorem 3.6 in (Munteanu et al., 2022) for ReLU; Theorem 1.1 in (Gao et al., 2023a) for exp; Corollary 4.3 in our paper for softmax.

|  | ReLU ((Munteanu et al., 2022)) | exp ((Gao et al., 2023a)) | Softmax (ours) |
|---|---|---|---|
| $m$ | $\Omega(\lambda^{-2}n^2\log(n))$ | $\Omega(\lambda^{-2}n^{2+o(1)}\log^2(n))$ | $\Omega(\lambda^{-2}n^{2+o(1)}\log^2(n))$ |
| $\widehat{T}$ | $\Omega(\lambda^{-2}n^2\log(n/\epsilon))$ | $\Omega(\lambda^{-2}n^{2+o(1)}\log(n/\epsilon))$ | $\Omega(\lambda^{-2}n^{2+o(1)}\log(n/\epsilon))$ |

input sequences by giving higher attention scores to the positions that are more relevant for the current position's representation and to capture dependencies between positions. (Cordonnier et al., 2020) find that softmax attention is more expressive and performs better than any convolutional layer. (Deng et al., 2023c) exhibits softmax attention outperforms linear attention in most scenarios. Although the softmax function code has been executed every second on thousands of servers, there is a limited understanding of the following question:

($*$) *What is the learning mechanism that makes softmax so powerful?*

To demystify the black box, in this paper, we analyze the Gradient Descent (GD) training dynamics for two-layer Neural Networks (NN) with softmax activation function for multi-dimensional regression, i.e., $F(W, x, a) \in \mathbb{R}^d$ and $\forall \ell \in \{1, \dots, d\}$,

$$F(W, x, a)_\ell := m\langle a_\ell, \exp(W^\top x)\rangle \cdot \langle \exp(W^\top x), \mathbf{1}_m\rangle^{-1},$$

where $m$ is number of hidden neurons, $\exp(\cdot)$ is element-wise exponential function, $a_\ell, W$ are the first and second layer weights respectively and $x$ is the input data. Note that, the self-attention could be written as $F(W^K X, W^Q X, W^V X) \in \mathbb{R}^{d \times n'}$, where $W^K, W^Q, W^V \in \mathbb{R}^{d \times d}$ denotes key, query, and value matrix and $X \in \mathbb{R}^{d \times n'}$ is a sequence of $n'$ tokens. Thus, studying the two-layer softmax network is the prerequisite to understanding self-attention. See more discussion in Section H.

There is a rich line of work studying two-layer NN learning trajectory under ReLU activation function ((Li & Liang, 2018; Du et al., 2019b; Allen-Zhu et al., 2019b; Arora et al., 2019a; Song & Yang, 2019; Mei et al., 2019; Song et al., 2021c; Brand et al., 2021; Munteanu et al., 2022; Chizat & Bach, 2020; Zhou et al., 2021; Lyu et al., 2021; Cao et al., 2022) and many more) or exponential activation function from the latest work (Gao et al., 2023a). As far as we know, our work is the first to theoretically study the optimization and generalization of the two-layer softmax network and it is a first step on understanding the power of softmax.

One popular analysis method for studying over-parameterized NN is Neural Tangent Kernel (NTK) (Jacot et al., 2018), where overparameterized networks are approximately linear models around their initialization, so the network training is almost convex.

To answer our ($*$) question above, we adopt the powerful NTK analysis paradigm in this work. Our analysis shows that, because of the normalization effect of the denominator, the Neural Tangent Kernel induced by the softmax has a good perturbation property (Lemma 5.1), which means the loss landscape of the softmax version has a large convex region. Thus, the softmax NN requires almost the same number of neurons and training steps to fit the data and converge as ReLU or exponential NN, which is illustrated in Table 1 clearly (Theorem 4.2). To demonstrate the broad applicability of our theoretical findings, we apply our analysis in a practical case study to show the generalization ability of softmax NN, where the task is learning score estimation functions in diffusion models with noisy labels, a promising approach for generative modeling, as we can smartly transfer it to a multi-dimensional regression task (Theorem 6.6). Thus, we show that gradient-based algorithms can learn the score function with a provable accuracy.

Our paper's contributions are summarized as follows:

- **Softmax NTK:** We build up the first NTK analysis framework for two-layer NN with softmax activation function (Theorem 4.2). Furthermore, our multi-dimensional regression setting is more general than previous work (Munteanu et al., 2022; Gao et al., 2023a) (ReLU and $\exp$) and can be degenerated to the linear regression setting.

- **Diffusion Models Case Study:** We apply our results in learning score estimation functions in diffusion models with noisy labels to verify our analysis effectiveness (Theorem 6.6).

## 2 RELATED WORKS

### 2.1 NEURAL TANGENT KERNEL

Recently many studies show that the analysis of optimization and generalization for deep learning should be interwoven together. One line of work uses the first-order Tyler expansion to study sufficiently over-parameterized neural networks around its initialization like NTK, e.g. (Matthews et al., 2018; Zou et al., 2018; Jacot et al., 2018; Li & Liang, 2018; Allen-Zhu et al., 2019c; Zou & Gu, 2019; Oymak & Soltanolkotabi, 2019; Lee et al., 2019; Novak et al., 2019; Yang, 2019; Song & Yang, 2019; Du et al., 2019a; Allen-Zhu et al., 2019b; Chizat et al., 2019; Oymak et al., 2019; Arora et al., 2019a; Cao & Gu, 2019; Ji & Telgarsky, 2019; Allen-Zhu et al., 2019a; Oymak & Soltanolkotabi, 2020; Cao et al., 2020; Zou et al., 2020; Geiger et al., 2020; Brand et al., 2021; Montanari & Zhong, 2022; Munteanu et al., 2022; Gao et al., 2023a; Qin et al., 2023b;a;c; Song & Ye, 2023; Gao et al., 2024; Song et al., 2024b) and more. Thus, the neural network optimization can be a convex problem. The NTK method has been widely used in different scenarios, such as preprocessing analysis (Song et al., 2021c; Hu et al., 2022; Alman et al., 2023; Shi et al., 2023a; Sun et al., 2023; 2024; Gao et al., 2024), federated learning (Li et al., 2023b), LoRA adaptation (Hu et al., 2021; Xu et al., 2024b; Shi et al., 2023b) of LLMs (Malladi et al., 2023), and learning score estimation functions in diffusion models (Han et al., 2024b).

### 2.2 SOFTMAX AND ATTENTION IN LLMS

Recently, significant advances have been achieved in language modeling, particularly with the introduction of Transformer architectures and attention mechanisms (Vaswani et al., 2017). Self-attention to capture long-range dependencies in text, revolutionizing the field of NLP, e.g., BERT (Devlin et al., 2019), PaLM (Chowdhery et al., 2022), LLaMA (Touvron et al., 2023a), LLaMA 2 (Touvron et al., 2023b), ChatGPT (OpenAI, 2022), GPT4 (Achiam et al., 2023), Claude 3 (Anthropic, 2024) and so on. Many works demonstrate the softmax is beyond other activation functions such as ReLU attention or linear attention in different aspects, e.g, approximation power (Deng et al., 2023c; Sanford et al., 2024; Noci et al., 2024; Li et al., 2024), prompt tuning (Oymak et al., 2023), in-context learning ability (Gao et al., 2023c; Shi et al., 2023c; Collins et al., 2024; Chen et al., 2024c), compositional ability(Xu et al., 2024a). Many works study to generalize the softmax into high order attention (Alman & Song, 2024b) or to accelerate softmax computation (Wang et al., 2020; Choromanski et al., 2020; Shen et al., 2021; Qin et al., 2021; Alman & Song, 2023; Brand et al., 2024; Alman & Song, 2024a; Han et al., 2024a; Hu et al., 2024; Deng et al., 2024; Song et al., 2024a; Gao et al., 2023d;e; Kacham et al., 2023; Liang et al., 2024b). Another line of work analyzes a one-layer softmax network trained on the linear regression task (Li et al., 2023a; Deng et al., 2023a;b; Chu et al., 2024; Gao et al., 2023b; Sheen et al., 2024), while our work studies a two-layer softmax setting.

### 2.3 DIFFUSION MODEL

Score-based generative diffusion models can generate high-quality image samples comparable to GANs which requires adversarial optimization (Ho et al., 2020; Song et al., 2021b; Kim et al., 2024). Based on the U-Net (Ronneberger et al., 2015), stable diffusion can successfully generate business-used images. Based on the softmax-based self-attention (Peebles & Xie, 2023), OpenAI released a video diffusion model, SORA (OpenAI, 2024), with a surprising performance. Another line of work study training diffusion models with a better theoretical guarantee (Song & Ermon, 2019; 2020; Song & Kingma, 2021; Song et al., 2020; 2021a; Lee et al., 2022; Kwon et al., 2022; Song et al., 2023; Lim et al., 2023; Chen et al., 2023a;d;b; Shah et al., 2023; Yang et al., 2023;

Benton et al., 2023; Gatmiry et al., 2024; Chen et al., 2024a; Guo et al., 2024; Wu et al., 2024; Chen et al., 2024b). In this work, we adapt our analysis in diffusion models.

**Roadmap.** We organize our paper as follows: In Section 3, we introduce the notation system and problem setup. In Section 4, we present our main result, proving that a Softmax neural network with $\text{poly}(nd)$ neurons can fit any training dataset consisting of $n$ $d$-dimensional samples for $d$-dimensional regression tasks. In Section 5, we outline the key techniques used to establish our main result. In Section 6, we extend our findings to Diffusion Models, demonstrating that Softmax neural networks can accurately learn score estimation even with noisy labels. Finally, in Section 7, we conclude the paper.

## 3 Preliminary

We first introduce some notations. Then, we will introduce our problem setup.

**Notations.** We use $\mathcal{N}(\mu, \Sigma)$ to denote the Gaussian distribution with $\mu$ and covariance $\Sigma$. For any positive integer $n$, we use $[n]$ to denote set $\{1, 2, \cdots, n\}$.

Let a vector $z \in \mathbb{R}^n$. We denote the $\ell_2$ norm as $\|z\|_2 := (\sum_{i=1}^n z_i^2)^{1/2}$, the $\ell_1$ norm as $\|z\|_1 := \sum_{i=1}^n |z_i|$, $\|z\|_0$ as the number of non-zero entries in $z$, $\|z\|_\infty$ as $\max_{i \in [n]} |z_i|$. We use $z^\top$ to denote the transpose of a $z$. We use $\langle \cdot, \cdot \rangle$ to denote the inner product. Let $A \in \mathbb{R}^{n \times d}$, we use $\text{vec}(A)$ to denote a length $nd$ vector. We denote the Frobenius norm as $\|A\|_F := (\sum_{i \in [n], j \in [d]} A_{i,j}^2)^{1/2}$. For a function $f(x)$, $f$ is $L$-Lipschitz if $\|f(x) - f(y)\|_2 \le L \cdot \|x - y\|_2$. Let $\mathcal{D}$ denote a distribution. We use $x \sim \mathcal{D}$ to denote that we sample a random variable $x$ from distribution $\mathcal{D}$. We use $\mathbb{E}[]$ to denote expectation and $\Pr[]$ to denote probability. We use p.s.d. to denote the positive-semidefinite matrix.

As we have multiple indexes, to avoid confusion, we usually use $i, j \in [n]$ to index the training data, $\ell \in [d]$ to index the output dimension, $r \in [m]$ to index neuron number.

### 3.1 Model, Data, and Algorithm

**Models.** We consider a two-layer softmax neural network. The hidden layer has $m$ neurons, and we use the softmax function as the activation function, $F(W, \cdot, a) : \mathbb{R}^{d_1} \to \mathbb{R}^{d_2}$ and $\forall \ell \in [d_2]$

$$F(W, x, a)_\ell := m \langle a_\ell, \exp(W^\top x) \rangle \cdot \langle \exp(W^\top x), \mathbf{1}_m \rangle^{-1}, \tag{1}$$

where $\exp(\cdot)$ is element-wise exponential function. We use $m$ as a normalization factor. Note that we can reduce the $d_2$ to 1 for the linear regression setting. To simplify the proof, we let $d_1 = d_2$. Note that our proof can generalize to different $d_1, d_2$ easily.

We only optimizing $W$ and not both $W$ and $a$ simultaneously as many previous works to simplify optimization, e.g., (Du et al., 2019b; Song & Yang, 2019; Munteanu et al., 2022), where $x \in \mathbb{R}^d$ represents the input, $w_1, \cdots, w_m \in \mathbb{R}^d$ are weight vectors in the first layer, i.e., $W = [w_1, \cdots, w_m] \in \mathbb{R}^{d \times m}$, and $a_1, \cdots, a_d \in \mathbb{R}^m$ are weights in the second layer. We can simplify the notation as $F(W, x)$ when the context is clear.

**Data.** We have $n$ training data points $\mathcal{D}_n = \{(x_i, y_i)\}_{i=1}^n$, where $x \in \mathbb{R}^d$ and $y \in \mathbb{R}^d$.[1] We denote $X = [x_1, \ldots, x_n] \in \mathbb{R}^{d \times n}$ and $Y = [y_1, \ldots, y_n] \in \mathbb{R}^{d \times n}$. We assume that $\|x_i\|_2 \le 1$ and $\|y_i\|_2 \le 1, \forall i \in [n]$.

**Gradient Descent.** We use $e_r$ to denote a vector where the $r$-th coordinate is 1 and everywhere else is 0. $\forall r \in [m], \forall \ell \in [d]$, we have $\frac{\partial F(W, x, a)_\ell}{\partial w_r} \in \mathbb{R}^d$ can be written as

$$
\begin{aligned}
\frac{\partial F(W, x, a)_\ell}{\partial w_r} &= + m \langle a_\ell \circ e_r, \exp(W^\top x) \rangle \cdot \langle \exp(W^\top x), \mathbf{1}_m \rangle^{-1} x \\
&\quad - m \langle a_\ell, \exp(W^\top x) \rangle \cdot \langle \exp(W^\top x), \mathbf{1}_m \rangle^{-2} \cdot \langle \exp(W^\top x), e_r \circ \mathbf{1}_m \rangle x \\
&= + m \langle a_\ell \circ e_r, \mathcal{S} \rangle \cdot x - m \langle a_\ell, \mathcal{S} \rangle \cdot \langle \mathcal{S}, e_r \circ \mathbf{1}_m \rangle x. \tag{2}
\end{aligned}
$$

---

[1] Our analysis can extend to $x_i \in \mathbb{R}^{d_1}$ and $y_i \in \mathbb{R}^{d_2}$ easily.

We have the softmax function $\mathcal{S} \in \mathbb{R}^{m \times n}$, where $\mathcal{S}_i \in \mathbb{R}^m$ denotes $\langle \exp(W^\top x_i), \mathbf{1}_m \rangle^{-1} \cdot \exp(W^\top x_i)$ and $\mathcal{S}_{i,r} \in \mathbb{R}$ denotes $\langle \exp(W^\top x_i), \mathbf{1}_m \rangle^{-1} \cdot \exp(w_r^\top x_i)$, $\forall r \in [m], \forall i \in [n]$. For simplicity, we denote $\alpha_i$ as $\langle \mathbf{1}_m, \exp(W^\top x_i) \rangle$, $\exp_i$ as $\exp(W^\top x_i)$ and $\exp_{i,r}$ as $\exp(w_r^\top x_i)$, $\forall r \in [m], \forall i \in [n]$, when the context is clear.

We use $W(\tau)$ to denote the weights of the first layer on the timestamp $\tau$ and similar for $\mathcal{S}(\tau)$ and $F(\tau)$ when the context is clear. Now, we introduce some necessary definitions used.

We first introduce the function over the whole training dynamic.

**Definition 3.1** ($F(\tau)$, dynamic prediction). *We define $F_i(\tau) \in \mathbb{R}^d$, for any timestamp $\tau$, as*

$$F_{\ell,i}(\tau) := m\langle a_\ell, \exp(W(\tau)^\top x_i) \rangle \cdot \langle \exp(W(\tau)^\top x_i), \mathbf{1}_m \rangle^{-1}.$$

*Here $x_i \in \mathbb{R}^d$. It can be rewritten as $F_{\ell,i}(\tau) = m\langle a_\ell, \mathcal{S}_i(\tau) \rangle$.*

We consider $d$-dimensional MSE loss.

**Definition 3.2** (Loss function over time). *We define the objective function $\mathcal{L}$ as below:*

$$\mathcal{L}(W(\tau)) := \frac{1}{2} \sum_{i \in [n]} \sum_{\ell \in [d]} (F_{\ell,i}(\tau) - y_{\ell,i})^2.$$

Thus, we define the gradient of $w$.

**Definition 3.3** ($\Delta w_r(\tau)$). *For any $r \in [m]$, we define $\Delta w_r(\tau) \in \mathbb{R}^d$ as below:*

$$\Delta w_r(\tau) := \frac{\mathrm{d}\mathcal{L}(W(\tau)}{\mathrm{d}w_r(\tau)}$$

$$= m \sum_{i=1}^n \sum_{\ell=1}^d (F_{\ell,i}(\tau) - y_{\ell,i}) \cdot \Big( \langle a_\ell \circ e_r, \mathcal{S}_i(\tau) \rangle - \langle a_\ell, \mathcal{S}_i(\tau) \rangle \cdot \langle \mathcal{S}_i(\tau), e_r \circ \mathbf{1}_m \rangle \Big) \cdot x_i$$

*where $\mathcal{S}_i(\tau) = \langle \exp(W(\tau)^\top x_i), \mathbf{1}_m \rangle^{-1} \cdot \exp(W(\tau)^\top x_i)$.*

We can simplify the gradient calculation by the fact $1 = \langle \mathbf{1}_m, \mathcal{S}_i(\tau) \rangle$. Thus, we have the following claim.

**Claim 3.4.** $\Delta w_r(\tau) := m \sum_{i=1}^n \sum_{\ell=1}^d (F_{\ell,i}(\tau) - y_{\ell,i}) \cdot \Big( (\langle a_{\ell,r} \cdot \mathbf{1}_m - a_\ell, \mathcal{S}_i(\tau) \rangle) \cdot \mathcal{S}_{i,r}(\tau) \Big) \cdot x_i.$

We use the gradient descent (GD) algorithm with the learning rate $\eta$ to train the network. As we only train the hidden layer $W$ and fix $a$, we have the following gradient update rule.

**Definition 3.5** (Gradient descent). *The gradient descent algorithm for optimizing the weight matrix $W$ is defined as:*

$$W(\tau + 1) = W(\tau) - \eta \Delta W(\tau),$$

*where $\Delta W(\tau) \in \mathbb{R}^{d \times m}$ and $\Delta w_r(\tau) \in \mathbb{R}^d$ is the $r$-th column of $\Delta W(\tau)$ defined in Definition 3.3.*

## 3.2 NEURAL TANGENT KERNEL

Now, we are ready to introduce our key tools, Neural Tangent Kernel induced by the softmax. We define the kernel with respect to timestamp $\tau$.

**Definition 3.6** (Kernel function). *For simplicity, we denote $\mathcal{S}(W^\top x_i)$ as $\mathcal{S}_i \in \mathbb{R}^m_{\geq 0}$ and $v_{\ell,r} = a_{\ell,r} \cdot \mathbf{1}_m - a_\ell \in \mathbb{R}^m$. We define the function (Gram matrix) $H : \mathbb{R}^{d \times m} \to \mathbb{R}^{nd \times nd}$ as following*

$$H(W) := \begin{bmatrix} H_{1,1} & H_{1,2} & \cdots & H_{1,d} \\ H_{2,1} & H_{2,2} & \cdots & H_{2,d} \\ \vdots & \vdots & \ddots & \vdots \\ H_{d,1} & H_{d,2} & \cdots & H_{d,d} \end{bmatrix},$$

*and for each $\ell_1, \ell_2 \in [d]$, we have $H_{\ell_1,\ell_2} \in \mathbb{R}^{n \times n}$ is defined as*

$$[H_{\ell_1,\ell_2}]_{i,j}(W) := \frac{1}{m} x_i^\top x_j \sum_{r=1}^m \langle v_{\ell_1,r}, \mathcal{S}_i \rangle \cdot m\mathcal{S}_{i,r} \cdot \langle v_{\ell_2,r}, \mathcal{S}_j \rangle \cdot m\mathcal{S}_{j,r}.$$

*For any timestamp $\tau$, for simplicity, we denote $H(\tau) := H(W(\tau))$ and denote $H(0)$ as $H^*$.*

Note that $H^*$ is a positive semi-definite matrix, and we denote its minimum eigenvalue as $\lambda := \lambda_{\min}(H^*)$ and we assume $\lambda > 0$ as previous works (Du et al., 2019b; Allen-Zhu et al., 2019b;c).

**Initialization.** We use symmetric initialization, which is widely used in previous works (Daniely & Malach, 2020; Damian et al., 2022; Munteanu et al., 2022; Shi et al., 2022; 2024).

**Definition 3.7** (Symmetric initialization). *For each $r \in [m/2]$, we initialize weights as below*

- *We draw $w_{2r-1}$ from $\mathcal{N}(0, \sigma^2 I_d)$ and uniformly draw $a_{2r-1}$ from $\{-1, +1\}^d$.*

- *We assign $a_{2r} = -a_{2r-1}$ and $w_{2r-1} = w_{2r}$.*

Due to symmetric initialization, we can easily see that $F(W(0), x) = 0, \forall x \in \mathbb{R}^d$.

## 4    MAIN RESULTS

We first define a constant we used.

**Definition 4.1.** *Let $C > 10$ denote a sufficiently large constant. We define parameter $B$ as follows $B := \max\{C\sigma\sqrt{\log(nd/\delta)}, 1\}$.*

Now, we are ready to present our main result, whose complete proof is in Appendix C.1.

**Theorem 4.2** (Main result). *Let $\lambda = \lambda_{\min}(H^*) > 0$, $m = \Omega(\lambda^{-2}n^2d^2 \exp(18B) \log^2(nd/\delta))$, $\eta = 0.1\lambda/(mn^2d^2 \exp(16B))$, and $\widehat{T} = \Omega((m\eta\lambda)^{-1} \log(nd/\epsilon)) = \Omega(\lambda^{-2}n^2d^2 \exp(16B) \cdot \log(nd/\epsilon))$. For any $\epsilon, \delta \in (0, 0.1)$, after $\widehat{T}$ iterations, with probability at least $1 - \delta$, we have $\|F(\widehat{T}) - Y\|_F^2 \le \epsilon$.*

If we fix $\delta$ and $\sigma$ in $B$ defined in the Definition 4.1, since $\exp(\Theta(B)) = (nd)^{o(1)}$, we can simplify the $m = \Omega(\lambda^{-2}(nd)^{2+o(1)})$ and $\widehat{T} = \Omega(\lambda^{-2}(nd)^{2+o(1)})$.

The Theorem 4.2 means that as we have $\text{poly}(nd)$ number of neurons and training steps, the softmax NN can fit any training datasets with $n$ number of $d$-dim training samples on $d$-dim regression task.

**Corollary 4.3.** *Consider the 1-dimension linear regression setting, i.e., $d_1 = d$ and $d_2 = 1$. Let $\lambda = \lambda_{\min}(H^*) > 0$, $m = \Omega(\lambda^{-2}n^2 \exp(18B) \log^2(n/\delta))$, $\eta = 0.1\lambda/(mn^2 \exp(16B))$, and $\widehat{T} = \Omega((m\eta\lambda)^{-1} \log(n/\epsilon)) = \Omega(\lambda^{-2}n^2 \exp(16B) \cdot \log(n/\epsilon))$. For any $\epsilon, \delta \in (0, 0.1)$, after $\widehat{T}$ iterations, with probability at least $1 - \delta$, we have $\|F(\widehat{T}) - Y\|_2^2 \le \epsilon$.*

*Proof.* Directly follow Theorem 4.2. $\qquad\square$

As shown in Table 1, our two-layer softmax network needs the same number of training steps $\widehat{T}$ and number of neurons $m$ as two-layer ReLU networks or two-layer exponential networks.

## 5    TECHNICAL OVERVIEW

We first show a key Lemma below, showing that the weight $w$ perturbation will not change the Neural Tangent Kernel too much.

**Lemma 5.1** (Weight value perturbation $\Rightarrow$ kernel value perturbation). *Let $R \in (0, 0.01)$. If the following conditions hold*

- *Let $\widetilde{W} = [\widetilde{w}_1, \cdots, \widetilde{w}_m] \in \mathbb{R}^{d \times m}$, where $\widetilde{w}_1, \cdots, \widetilde{w}_m$ are i.i.d. draw from $\mathcal{N}(0, \sigma^2 I_d)$.*

- *Let $W = [w_1, \cdots, w_m] \in \mathbb{R}^{d \times m}$ and satisfy $\|\widetilde{w}_r - w_r\|_2 \le R$ for any $r \in [m]$.*

*Then, with probability at least $1 - \delta$, we have $\|H(W) - H(\widetilde{W})\|_F \le Rnd \exp(10B)$.*

Please see Appendix B.2 for the proof of Lemma 5.1. We can see that the kernel matrix has a small perturbation when the weights $w$ perturb. Note that in Lemma 4.2 of (Munteanu et al., 2022), they have $\|H(W) - H(\widetilde{W})\|_F \le 2Rn$ for the ReLU activation function and in Lemma 6.7 of (Gao et al.,

2023a), they have $\|H(W) - H(\widetilde{W})\|_F \leq 3Rn^{1+o(1)}$ for the $\exp$ activation function. When we consider the 1-dimension linear regression task, we have $\|H(W) - H(\widetilde{W})\|_F \leq Rn^{1+o(1)}$, which is almost the same as the other two cases.

**Remark 5.2.** *In the proof of Lemma B.2, we do not use concentration bound as previous work (Song & Yang, 2019; Munteanu et al., 2022; Gao et al., 2023a). The reason is that we consider the worst case. In general, $\mathbb{E}[H(W) - H(\widetilde{W})] \neq \mathbf{0}_{nd \times nd}$. Thus, using the concentration bound may not gain any benefits.*

Based on Lemma 5.1, we can use math induction to finish the proof of our main Theorem. We show the induction statement below.

**Lemma 5.3** (Induction). *Let $\tau$ be a fixed integer. Assume the same condition as Theorem 4.2. Let $D$ be defined as Definition A.2 and $D < R$. If the following conditions hold*

- **Weights Property.** $\|w_r(i) - w_r(0)\|_2 \leq R$, $\forall i \in [\tau]$
- **Loss Property.** $\|F(i) - Y\|_F^2 \leq \|F(0) - Y\|_F^2 \cdot (1 - m\eta\lambda/2)^i$, $\forall i \in [\tau]$
- **Gradient Property.** $\eta\|\Delta w_r(i)\|_2 \leq 0.01$ *for all* $r \in [m]$, $\forall i \in [\tau]$

*Then, for $\tau + 1$ and $\forall r \in [m]$, we have*

- **Weights Induction.** $\|w_r(\tau + 1) - w_r(0)\|_2 \leq D$.
- **Loss Induction.** $\|F(\tau + 1) - Y\|_F^2 \leq (1 - m\eta\lambda/4)^{\tau+1} \cdot \|F(0) - Y\|_F^2$.
- **Gradient Induction.** $\eta\|\Delta w_r(\tau + 1)\|_2 \leq 0.01$, $\forall r \in [m]$.

Please refer to Appendix C.2, Appendix C.3 and Appendix C.4 for the proof of weights, loss, gradient induction in Lemma 5.3 respectively.

Lemma 5.3 means that, at a fixed timestamp $\tau$, if the weights $w(\tau)$ is close to its initialization, the loss is decreasing, and the gradient is also small, then we can conclude at timestamp $\tau + 1$, these conditions still hold as local convexity proved by Lemma 5.1. Thus, after checking the initial condition, we can conclude Theorem 4.2.

## 5.1 TECHNICAL NOVELTY AND COMPARISON TO THE EXISTING LITERATURE

In this work, as we consider the softmax activation function, the denominator term will also contribute to gradient calculation. Handling the denominator poses many technical challenges, where these challenges are unique to our setting and not presented in previous settings as ReLU (Song & Yang, 2019), or $\exp$ (Gao et al., 2023a) activation function. In detail, in the gradient calculation, we need new loss decomposition Lemma E.1 to split the loss into $\|F(\tau + 1) - Y\|_F^2 = \|F(t) - Y\|_F^2 + C_0 + C_1 + C_2 + C_3$. Then, we need to bound these new terms in Lemma E.3 for $C_0$, Lemma E.4 and Claim E.5 for $C_1$, Claim E.6 for $C_2$ and Claim E.7 for $C_3$, where all these Lemmas are novel and non-trivial. We refer readers to Appendix C.3 for more details.

## 6 EXTENSION ON DIFFUSION

Now, we apply our results in learning score estimation functions in diffusion models with noisy labels. We introduce problem setup in Section 6.1 and show our results in Section 6.2.

## 6.1 PRELIMINARY OF DIFFUSION

In this section, we briefly introduce the diffusion model proposed in (Song et al., 2021b).

**Forward Process.** During the forward process, we progressively inject the noise into the original data distribution, which can be characterized by the following Stochastic Differential Equation (SDE) (Song & Ermon, 2020; Ho et al., 2020):

$$\mathrm{d}x(t) = -\frac{1}{2}g(t)x(t)\,\mathrm{d}t + \sqrt{g(t)}\mathrm{d}B_t, \quad x(0) \sim p_0, \tag{3}$$

where $x(t)$ is the data at the diffusion process time $t$, $g(t) > 0$ is a deterministic weighting function; and $(B_t)_{t \geq 0}$ is a standard $d$-dimensional Brownian motion/noise. The $p_0$ represents the original/target data distribution that we learn, and we only have few number of accesses to it, i.e., $n$ times. We denote $p_t$ as the distribution of $x(t)$ at diffusion process time $t$. Then, we can write the explicit solution to Eq. (3) as

$$x(t) = e^{-\int_0^t \frac{1}{2}g(s)\mathrm{d}s} x(0) + e^{-\int_0^t \frac{1}{2}g(s)\mathrm{d}s} \int_0^t e^{\int_0^s \frac{1}{2}g(u)\mathrm{d}u} \sqrt{g(s)}\mathrm{d}B_s.$$

**Backward Process.** We denote $y(t) = x(T - t)$ to reverse the forward process in time (Haussmann & Pardoux, 1986; Föllmer, 2005; Cattiaux et al., 2021) that transforms noise into samples from the target distribution. We have a backward process associated to Eq. (3) as:

$$\mathrm{d}y(t) = (\frac{1}{2}g(T - t)y(t) + g(T - t)\nabla \log p_{T-t}(y(t)))\mathrm{d} + \sqrt{g(T - t)}\mathrm{d}\bar{B}_t, \quad y(0) \sim q_0. \quad (4)$$

where $(\bar{B}_t)_{t \geq 0}$ is another $d$-dim Brownian motion/noise. Following the literature, we call $\nabla \log p_t(\cdot)$ as "score function" (Song et al., 2021b). We have $q_0$ as the initial distribution of the backward process and the score function $\nabla \log p_t(\cdot)$ as the gradient of log density of $x(t)$.

However, In practice, Eq.(4) cannot be directly used as both the score function and the distribution $p_T$ are unknown. To solve the problem, we (1) randomly select a noise distribution as the initial distribution of the backward process $p_T$; (2) replace the ground-truth score function $\nabla \log p_t(x(t))$ by an estimator $s_\theta(x(t), t)$. The parameterized estimator $s_\theta$ is learned by a neural network such as U-Net (Ho et al., 2020; Rombach et al., 2022) and Transformer (Peebles & Xie, 2023). Thus, we obtain a practically implementable approximation of the backward SDE:

$$\mathrm{d}y(t) = (\frac{1}{2}g(T - t)y(t) + g(T - t)s_\theta(y(t), t))\mathrm{d}t + \sqrt{g(T - t)}\mathrm{d}\bar{B}_t, \quad y(0) \sim \mathcal{N}(0, I_d),$$

which can be used for sampling/data generation (Song & Ermon, 2020; Chen et al., 2023b;c)

**Score Matching.** When estimating the score function, we usually use $L_2$ loss between the estimated and actual score:

$$\min_\theta \frac{1}{T} \int_0^T \lambda(t)\mathbb{E}[\|s_\theta(x(t), t) - \nabla \log p_t(x(t))\|_2^2]\mathrm{d}t, \quad (5)$$

where $\lambda(t)$ is the weighting function that captures time inhomogeneity. As the hardness of estimate $\nabla \log p_t$ term in Eq. (5), equivalently, we minimize the following denoising score matching (Vincent, 2011):

$$\min_\theta \frac{1}{T - T_0} \cdot \int_{T_0}^T \lambda(t)\mathbb{E}[\|s_\theta(x(t), t) - \nabla \log p_{t|0}(x(t) \mid x(0))\|_2^2]\mathrm{d}t. \quad (6)$$

In practice, the estimator of the score function is parameterized by a neural network, and we have the following sampling procedure for any $i \in [n]$,

$$x(0)_i \sim p_0, \quad t_i \sim \mathrm{Unif}(0, T), \quad x(t_i)_i \sim p_{t_i|0}(\cdot | x(0)_i),$$

and we get the training dataset $\{x(0)_i, (t_i, x(t_i)_i)\}_{i=1}^n$, where $x(0)_i \in \mathbb{R}^d$ and $(t_i, x(t_i)_i) \in \mathbb{R}^{d+1}$. We denote $x(0)$ as the noisy label and $\mathbb{E}[x(0)|x(t)]$ as the true label. For simplicity, we denote $x(0)_i$ as $y_i \in \mathbb{R}^d$ and $(t_i, x(t_i)_i)$ as $x_i \in \mathbb{R}^{d+1}$ and the training dataset as $\mathcal{D}_n = \{(x_i, y_i)\}_{i=1}^n$. Here, $y$ denotes the image from a dataset, and $x$ denotes the noised image with its diffusion process time $t$.

**Neural Network Parameterization.** Recall that we consider a two-layer network with softmax activation function as the diffusion model in Eq. (1), satisfying $\forall \ell \in [d]$, $F(W, x, a)_\ell = m\langle a_\ell, \exp(W^\top x)\rangle \cdot \langle \exp(W^\top x), \mathbf{1}_m\rangle^{-1}$. Note that we do not train the top-layer weights $a$, so we can denote it as $F_{nn}(W, x)$.

Then, similar as (Ho et al., 2020; Han et al., 2024b), our loss function Eq. (6) can be rewrite as

$$\min_W \mathcal{L}(W) := \frac{1}{2} \sum_{j=1}^N \|F_{nn}(W, x_j) - y_j\|_2^2.$$

We denote the target function as $F_*(t, x(t)) := \mathbb{E}[y \mid (t, x(t))]$. Let $\mathcal{H}$ be the reproducing Hilbert space (RKHS) induced by the NTK (Carmeli et al., 2010; Jacot et al., 2018) and let $F_\mathcal{H}$ in the RKHS $\mathcal{H}$ such that $\|F_\mathcal{H}\|_\mathcal{H}^2 \leq R_\mathcal{H}$.

## 6.2 MAIN RESULT OF DIFFUSION

We first introduce some natural assumptions we used.

**Assumption 6.1.** *Based on normalization, we assume $\|y_i\|_2 \leq 1, \|x_i\|_2 \leq 1, \forall i \in [n]$.*

**Assumption 6.2.** *Assume $\lambda = \lambda_{\min}(H^*) > 0$.*

**Assumption 6.3.** *The function $g$ is almost everywhere continuous and bounded on $[0, \infty)$.*

**Assumption 6.4.** *For all $(t, x(t)) \in (0, \infty) \times \mathbb{R}^d$, the function $F_*(t, x(t))$ is $\beta_x$-Lipschitz in $x$, i.e., $\|F_*(t, x(t)) - F_*(t, x'(t))\|_2 \leq \beta_x \|x(t) - x'(t)\|_2$.*

We denote $A(R_{\mathcal{H}}) := c_1 \Lambda (\frac{\sqrt{R_{\mathcal{H}}}}{\Lambda})^{-\frac{2}{d}} \log(\frac{\sqrt{R_{\mathcal{H}}}}{\Lambda})$ and $\Lambda = O(\sqrt{d})$ and

$$\Gamma_\delta := \left( \frac{2d^2 A(R_{\mathcal{H}})}{\lambda} \log^{3/2}(\frac{e(dn)^{3/2} A(R_{\mathcal{H}})}{\lambda}) + \frac{1}{\sqrt{n}} \right)^2 + \frac{d^2 A^2(R_{\mathcal{H}})}{\lambda^2} (\log(1/\delta) + \log(\log n)).$$

**Assumption 6.5** (Assumption 3.11 in (Han et al., 2024b)). *Fix any $F_{\mathcal{H}} \in \mathcal{H}$ with $\|F_{\mathcal{H}}\|_{\mathcal{H}}^2 \leq R_{\mathcal{H}}$ and assume labels are generated as $\widetilde{y}_j = F_{\mathcal{H}}(x_j) + \epsilon_j$. Suppose $\widetilde{F}_{ntk}(\gamma(\widehat{T}), \cdot)$ is obtained by GD-trained kernel regression with the number of iterations $\widehat{T}$. We assume there exists $\epsilon$ such that*

$$\frac{1}{T} \int_0^T \mathbb{E}[\widetilde{F}_{ntk}(\gamma(\widehat{T}), (t, x(t))) - F_{\mathcal{H}}(t, x(t))\|_2^2] \mathrm{d}t \leq \epsilon(n, \widehat{T}),$$

*and $\epsilon(n, \widehat{T}) \to 0$ as $n \to \infty$.*

Now, we are ready to present our main Theorem for diffusion.

**Theorem 6.6** (Main results of score estimation and generalization). *Suppose Assumptions 6.1, 6.2, 6.3, 6.4 hold and we set $m = \Omega(\lambda^{-2} n^3 d^3 \exp(18B) \log^2(nd/\delta))$ and $\eta = 0.1\lambda/(mn^2 d^2 \exp(16B))$. Moreover, suppose early stopping $\widehat{T}$ satisfies Assumption 6.5 with corresponding $\epsilon(n, \widehat{T})$. Then for large enough $R_{\mathcal{H}}$, with probability at least $1 - \delta$, it holds that*

$$\frac{1}{T} \int_0^T \lambda(t) \mathbb{E}[\|s_{W(\widehat{T})}(t, x(t)) - \nabla \log p_t(X_t)\|_2^2] \mathrm{d}t$$

$$\leq O(\frac{1}{\lambda\sqrt{n}} + \epsilon(n, \widehat{T}) + dA^2(R_{\mathcal{H}}) + dA(R_{\mathcal{H}}) + \sqrt{dA(R_{\mathcal{H}})\Gamma_\delta} + \Gamma_\delta).$$

Please refer to Appendix G.1 for the complete proof. Here, we provide a proof sketch.

*Proof sketch of Theorem 6.6.* In Theorem F.2, we show the "equivalence" between softmax NN learning and corresponding neural tangent kernel regression, i.e., the gap between them is always small. Then, we can borrow the generalization ability of kernel regression to the generalization ability of two-layer softmax NN. On the other hand, by Claim G.1, we can decompose the loss into a coupling gap, a label mismatch gap, an early stopping gap, and an approximation gap. By using our Theorem 4.2, Theorem F.2 with some tools from (Han et al., 2024b), we finish the proof. □

From Theorem 6.6, we know that, under some natural assumptions, the GD algorithm trained two-layer softmax NN can learn a provable accuracy on the score estimation functions in the diffusion model with noisy labels. We use this practical case study to demonstrate the broad applicability of our theoretical findings.

## 7 CONCLUSION

This paper provides a theoretical analysis of the optimization and generalization properties of two-layer neural networks with the softmax activation function. We apply our results in learning score estimation functions in diffusion models with noisy labels to verify our analysis effectiveness. Our findings contribute to a deeper understanding of the power of softmax neural networks and their potential for self-attention, advanced LLMs, and generative modeling.

## ETHIC STATEMENT

This paper does not involve human subjects, personally identifiable data, or sensitive applications. We do not foresee direct ethical risks. We follow the ICLR Code of Ethics and affirm that all aspects of this research comply with the principles of fairness, transparency, and integrity.

## REPRODUCIBILITY STATEMENT

We ensure reproducibility of our theoretical results by including all formal assumptions, definitions, and complete proofs in the appendix. The main text states each theorem clearly and refers to the detailed proofs. No external data or software is required.

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

# Appendix

**Roadmap.** In Section A, we introduce some definitions that will be used in the proof. In Section B, we provide the basic concentration. In Section C, we provide the proof of our inductions. In Section D, we establish a bound for the weight of induction Part 1. In Section E, we establish a bound for the loss of induction Part 2. In Section F, we introduce the NTK regression. In Section G, we introduce the diffusion. In Section H, we discuss the potential implications of our results for popular frameworks such as attention mechanisms and feature learning. In Section **??**, we provide the potential limitations of this work. In Section **??**, we discuss the societal impacts of our work.

## A  DEFINITION

**Claim A.1** (Restatement of Claim 3.4). *We have*

$$\Delta w_r(\tau) := m \sum_{i=1}^{n} \sum_{\ell=1}^{d} (F_{\ell,i}(\tau) - y_{\ell,i}) \cdot \Big( (\langle a_{\ell,r} \cdot \mathbf{1}_m - a_\ell, \mathcal{S}_i(\tau) \rangle) \cdot \mathcal{S}_{i,r}(\tau) \Big) \cdot x_i$$

*Proof of Claim 3.4.* We can show that

$$\Delta w_r(\tau)/m = \sum_{i=1}^{n} \sum_{\ell=1}^{d} (F_{\ell,i}(\tau) - y_{\ell,i}) \cdot (\langle a_\ell \circ e_r - a_\ell \cdot \mathcal{S}_{i,r}(\tau), \mathcal{S}_i(\tau) \rangle) x_i$$

$$= \sum_{i=1}^{n} \sum_{\ell=1}^{d} (F_{\ell,i}(\tau) - y_{\ell,i}) \cdot \Big( (a_{\ell,r} - \langle a_\ell, \mathcal{S}_i(\tau) \rangle) \cdot \mathcal{S}_{i,r}(\tau) \Big) \cdot x_i$$

$$= \sum_{i=1}^{n} \sum_{\ell=1}^{d} (F_{\ell,i}(\tau) - y_{\ell,i}) \cdot \Big( \underbrace{\langle a_{\ell,r} \cdot \mathbf{1}_m - a_\ell}_{m \times 1}, \underbrace{\mathcal{S}_i(\tau)}_{m \times 1} \rangle \cdot \mathcal{S}_{i,r}(\tau) \Big) \cdot x_i,$$

where the first step follows from the definition of $\Delta w_r(\tau)$, the second step follows from $\langle a_\ell \circ e_r, x \rangle = a_{\ell,r} x_r$, and the last step is due to the Fact A.4. $\qquad\square$

We present the following definition to simplify the notation.

**Definition A.2.** *We define $D$*

$$D := 4m^{-1}\lambda^{-1} \exp(3B)\sqrt{nd} \cdot \|F(0) - Y\|_F$$

**Fact A.3.** *For any vectors $u, v \in \mathbb{R}^n$, the squared Euclidean distance between $u$ and $v$ can be expressed as:*

$$\|u - v\|_2^2 = \|u\|_2^2 - 2u^\top v + \|v\|_2^2.$$

**Fact A.4.** *Let $\mathbf{1}_m$ be a vector of dimension $m$ consisting of all ones, and $\mathcal{S}_i(\tau) \in \mathbb{R}_{\geq 0}^m$ be the indicator of some function $\tau$ at position $i$. We have:*

$$1 = \langle \mathbf{1}_m, \mathcal{S}_i(\tau) \rangle$$

**Fact A.5.** *For any real number $|x| \leq 0.1$, the following inequality holds:*

$$(1 - x)^{1/2} \leq 1 - 0.5x$$

**Fact A.6.** *For any real number $|x| \leq 0.1$, we have*

$$|\exp(x) - 1| \leq 2|x|$$

**Fact A.7.** *For any $x \in (0, 0.1)$, we have*

$$\sum_{i=0}^{\infty} x^i \leq \frac{1}{1 - x}$$

**Fact A.8.** *For any* $|x| \leq 0.01$, *we have*

$$\exp(x) = 1 + x + \Theta(1)x^2$$

We state the standard Hoeffding inequality,

**Lemma A.9** (Hoeffding inequality (Hoeffding, 1963))**.** *If the below conditions are true*

- *Let* $x_1, \cdots, x_n$ *denote* $n$ *independent variables*

- $x_i \in [\alpha_i, \beta_i]$, *for all* $i \in [n]$

- *Let* $x = \sum_{i=1}^{n} x_i$.

*Then we have*

$$\Pr[|x - \mathbb{E}[x]| \geq t] \leq 2 \exp\left(-\frac{2t^2}{\sum_{i \in [n]} (\beta_i - \alpha_i)^2}\right).$$

**Lemma A.10** (Hanson-Wright inequality (Hanson & Wright, 1971; Rudelson & Vershynin, 2013))**.** *Let* $x \in \mathbb{R}^n$ *denote a random vector with independent entries* $x_i$ *with* $\mathbb{E}[x_i] = 0$ *and* $|x_i| \leq K$. *Let* $A$ *be an* $n \times n$ *matrix. Then, for every* $t \geq 0$,

$$\Pr[|x^\top A x - \mathbb{E}[x^\top A x]| > t] \leq 2 \cdot \exp(-c \min\{t^2/(K^4 \|A\|_F^2), t/(K^2 \|A\|)\}).$$

## B    BASIC CONCENTRATION

In Section B.1, we introduce some concentration basic tools. In Section B.2, given $w$ perturbation within a small ball, we bound the changes of $H$.

### B.1    SOME CONCENTRATION BASIC TOOLS

The goal of this section is to prove Lemma B.1.

**Lemma B.1.** *If the following conditions hold*

- *Let* $B > 1$ *denote a parameter be defined as Definition 4.1.*

- *Let* $W = [w_1, \cdots, w_m]$ *and* $w_r$ *be random Gaussian vectors from* $\mathcal{N}(0, \sigma^2 I_d)$.

- *Let* $V = [v_1, \cdots, v_m]$ *and* $v_r$ *denote the vector where* $\|v_r - w_r\|_2 \leq R$, $\forall r \in [m]$.

- *Let* $x_i \in \mathbb{R}^d$ *and* $\|x_i\|_2 \leq 1$, $\forall i \in [n]$.

- *Let* $R \in (0, 0.01)$.

- *Let* $\mathcal{S}_i$ *and* $\widetilde{\mathcal{S}}_i$ *be the softmax function corresponding to* $W$ *and* $V$ *respectively.*

- *Let* $\alpha_i = \langle \mathbf{1}_m, \exp(W^\top x_i) \rangle$ *and* $\widetilde{\alpha}_i = \langle \mathbf{1}_m, \exp(V^\top x_i) \rangle$, $\forall i \in [n]$.

*Then, with probability at least* $1 - \delta/\operatorname{poly}(nd)$, *we have*

- *Standard inner product*

  - *Part 1.* $|\langle w_r, x_i \rangle| \leq B$, $\forall i \in [n]$, $\forall r \in [m]$
  - *Part 2.* $|\langle v_r, x_i \rangle| \leq B + R$, $\forall i \in [n]$, $\forall r \in [m]$
  - *Part 3.* $|\langle w_r - v_r, x_i + x_j \rangle| \leq 2R$, $\forall i, j \in [n]$, $\forall r \in [m]$

- $\exp$ *function*

  - *Part 4.* $\exp(-B) \leq \exp(\langle w_r, x_i \rangle) \leq \exp(B)$, $\forall i \in [n]$, $\forall r \in [m]$
  - *Part 5.* $\exp(-B - R) \leq \exp(\langle v_r, x_i \rangle) \leq \exp(B + R)$, $\forall i \in [n]$, $\forall r \in [m]$
  - *Part 6.* $|\exp(\langle w_r - v_r, x_i + x_j \rangle) - 1| \leq 4R$, $\forall i, j \in [n]$, $\forall r \in [m]$

- **Part 7.** $|\exp(\langle w_r, x_i \rangle) - \exp(\langle v_r, x_i \rangle)| \leq R \exp(B + R), \forall i \in [n], \forall r \in [m]$

- *softmax $\mathcal{S}$ function*

  - **Part 8.** $|\alpha_i - \widetilde{\alpha}_i| \leq mR \exp(B + R), \forall i \in [n]$
  - **Part 9.** $|\alpha_i^{-1} - \widetilde{\alpha}_i^{-1}| \leq \frac{R}{m} \exp(3B + 2R), \forall i \in [n]$
  - **Part 10.** $|\mathcal{S}_{i,r}| \leq \exp(2B)/m, \forall i \in [n], \forall r \in [m]$
  - **Part 11.** $|\widetilde{\mathcal{S}}_{i,r}| \leq \exp(2B + 2R)/m, \forall i \in [n], \forall r \in [m]$
  - **Part 12.** $|\mathcal{S}_{i,r} - \widetilde{\mathcal{S}}_{i,r}| \leq \frac{R}{m} \exp(4B + 3R), \forall i \in [n], \forall r \in [m]$
  - **Part 13.** *for any $z \in \mathbb{R}^m$ and $\|z\|_\infty \leq 1$, we have* $|\langle z, \mathcal{S}_i \rangle - \langle z, \widetilde{\mathcal{S}}_i \rangle| \leq R \exp(4B + 3R), \forall i \in [n]$

*Proof.* As eventually we choose $m = \mathrm{poly}(nd)$, we use $B > 0$ defined in Definition 4.1.

**Proof of Part 1, 2, 4 and 5.**

We can get the proof by Gaussian tail bound.

**Proof of Part 3 and 6.**

Due to $\|x_i\|_2 \leq 1$ and $\|x_j\|_2 \leq 1$ and $\|\Delta w_r\|_2 \leq R$, we can have

$$|\langle \Delta w_r, (x_i + x_j) \rangle| \leq 2R \leq 0.1. \tag{7}$$

Then, we have

$$|\exp(\langle \Delta w_r, (x_i + x_j) \rangle) - 1| \leq 2|\langle \Delta w_r, (x_i + x_j) \rangle|$$
$$\leq 4R$$

where the first step follows from the Fact A.6, and the last step follows from Eq. (7).

**Proof of Part 7.** Because $\|x_i\|_2 \leq 1$ and $\|\Delta w_r\|_2 \leq R$, we can have

$$|\langle \Delta w_r, x_i \rangle| \leq R \leq 0.1. \tag{8}$$

By convex increasing property of $\exp$ function, we have

$$|\exp(\langle w_r, x_i \rangle) - \exp(\langle v_r, x_i \rangle)| \leq \max\{\exp'(\langle w_r, x_i \rangle), \exp'(\langle v_r, x_i \rangle)\} \cdot |\langle \Delta w_r, x_i \rangle|$$
$$\leq \exp(B + R) \cdot |\langle \Delta w_r, x_i \rangle|$$
$$\leq \exp(B + R)R.$$

where the first step follows from Taylor expansion and $\exp'$ denote the derivative of $\exp$, the second step follows from Part 4 and Part 5 and the last step follows from Eq. (8).

**Proof of Part 8.**

$$|\alpha_i - \widetilde{\alpha}_i| = |\sum_{r \in [m]} \exp_{i,r} - \widetilde{\sum_{r \in [m]} \exp_{i,r}}|$$
$$\leq \sum_{r \in [m]} |\exp_{i,r} - \widetilde{\exp}_{i,r}|$$
$$\leq mR \exp(B + R),$$

where the third step is due to Part 7.

**Proof of Part 9.**

Similarly, we have

$$|\alpha_i^{-1} - \widetilde{\alpha}_i^{-1}| = |\frac{\widetilde{\alpha}_i - \alpha_i}{\alpha_i \widetilde{\alpha}_i}|$$

$$\leq \frac{mR\exp(B+R)}{|\alpha_i \widetilde{\alpha}_i|}$$

$$\leq \frac{mR\exp(B+R)}{|m\exp(-B)m\exp(-B-R)|}$$

$$= \frac{R}{m}\exp(3B+2R).$$

where the first step is due to simple algebra, the second step is from Part 8, the third step follows Part 4, 5, and the last step is because of simple algebra.

**Proof of Part 10 and 11.**

Trivially follows Part 4 and Part 5.

**Proof of Part 12.**

$$|\mathcal{S}_{i,r} - \widetilde{S}_{i,r}| = |\alpha_i^{-1}\exp_{i,r} - \widetilde{\alpha}_i^{-1}\widetilde{\exp}_{i,r}|$$

$$\leq |\alpha_i^{-1}\exp_{i,r} - \alpha_i^{-1}\widetilde{\exp}_{i,r}| + |\alpha_i^{-1}\widetilde{\exp}_{i,r} - \widetilde{\alpha}_i^{-1}\widetilde{\exp}_{i,r}|$$

For the first part, we have

$$|\alpha_i^{-1}\exp_{i,r} - \alpha_i^{-1}\widetilde{\exp}_{i,r}| = \alpha_i^{-1}|\exp_{i,r} - \widetilde{\exp}_{i,r}|$$

$$\leq \alpha_i^{-1}\exp(B+R)R$$

$$\leq \frac{\exp(B+R)R}{m\exp(-B)}$$

$$= \frac{R}{m}\exp(2B+R),$$

where the second step follows Part 7 and the third step follows Part 4.

For the second part, we have

$$|\alpha_i^{-1}\widetilde{\exp}_{i,r} - \widetilde{\alpha}_i^{-1}\widetilde{\exp}_{i,r}| = \widetilde{\exp}_{i,r}|\alpha_i^{-1} - \widetilde{\alpha}_i^{-1}|$$

$$\leq \widetilde{\exp}_{i,r}\frac{R}{m}\exp(3B+2R)$$

$$\leq \exp(B+R)\frac{R}{m}\exp(3B+2R)$$

$$= \frac{R}{m}\exp(4B+3R),$$

where the second step follows Part 9, and the third step follows Part 5.

Thus, we have

$$|\mathcal{S}_{i,r} - \widetilde{S}_{i,r}| \leq \frac{R}{m}\exp(4B+3R).$$

**Proof of Part 13.**

Note that $\|z\|_\infty \leq 1$. We have

$$|\langle z, \mathcal{S}_i\rangle - \langle z, \widetilde{\mathcal{S}}_i\rangle| = |\langle z, \mathcal{S}_i - \widetilde{\mathcal{S}}_i\rangle|$$

$$\leq m\|\mathcal{S}_i - \widetilde{\mathcal{S}}_i\|_\infty$$

$$\leq R\exp(4B+3R)$$

where the first step follows from simple algebra, the second step follows from $|\langle a, b\rangle| \leq m \cdot \max_{i\in[m]}|a_i b_i|$, and the last step is due to Part 12. $\square$

## B.2 KERNEL PERTURBATION

The purpose of this section is to prove Lemma B.2. In the proof, we do not use concentration inequality. Please see Remark 5.2 for more details.

**Lemma B.2** (Restatement of Lemma 5.1). *If the following conditions hold*

- *Let $B \geq 1$ denote a parameter be defined as Definition 4.1.*

- *Let $R \in (0, 0.01)$.*

- *Let $x_i \in \mathbb{R}^d$ and $\|x_i\|_2 \leq 1$ for all $i \in [n]$.*

- *Let $\widetilde{W} = [\widetilde{w}_1, \cdots, \widetilde{w}_m] \in \mathbb{R}^{d \times m}$, where $\widetilde{w}_1, \cdots, \widetilde{w}_m$ are are i.i.d. draw from $\mathcal{N}(0, \sigma^2 I_d)$.*

- *Let $W = [w_1, \cdots, w_m] \in \mathbb{R}^{d \times m}$ and satisfy $\|\widetilde{w}_r - w_r\|_2 \leq R$ for any $r \in [m]$.*

- *Let $v_{\ell,r} = a_{\ell,r} \cdot \mathbf{1}_m - a_\ell \in \mathbb{R}^m$, for any $\ell \in [d]$ and for any $r \in [m]$. Note that $a_{\ell,r}$ is the $r$-th in $a_\ell$.*

- *Let $\alpha_i = \langle \mathbf{1}_m, \exp(W^\top x_i) \rangle$ and $\widetilde{\alpha}_i = \langle \mathbf{1}_m, \exp(V^\top x_i) \rangle$, $\forall i \in [n]$.*

- *Let $H$ be defined as Definition 3.6.*

*Then, we have*

- *Part 1. Then with probability at least $1 - \delta/\operatorname{poly}(nd)$,*
$$|[H_{\ell_1,\ell_2}]_{i,j}(W) - [H_{\ell_1,\ell_2}]_{i,j}(\widetilde{W})| \leq R \cdot \exp(10B).$$

- *Part 2. Then with probability at least $1 - \delta$, we have*
$$\|H(W) - H(\widetilde{W})\|_F \leq Rnd \cdot \exp(10B).$$

*Proof of Lemma 5.1.* We define five real numbers $B_1, B_2, B_3, B_4, B_5 \in \mathbb{R}$ as follows,

$$B_1 := \alpha_i^{-1}\alpha_j^{-1}\frac{1}{m}\sum_{r=1}^m \langle v_{\ell_1,r}, \mathcal{S}_i \rangle \langle v_{\ell_2,r}, \mathcal{S}_j \rangle \exp_{i,r}\exp_{j,r} - \alpha_i^{-1}\alpha_j^{-1}\frac{1}{m}\sum_{r=1}^m \langle v_{\ell_1,r}, \mathcal{S}_i \rangle \langle v_{\ell_2,r}, \mathcal{S}_j \rangle \widetilde{\exp}_{i,r}\widetilde{\exp}_{j,r}$$

$$B_2 := \alpha_i^{-1}\alpha_j^{-1}\frac{1}{m}\sum_{r=1}^m \langle v_{\ell_1,r}, \mathcal{S}_i \rangle \langle v_{\ell_2,r}, \mathcal{S}_j \rangle \widetilde{\exp}_{i,r}\widetilde{\exp}_{j,r} - \alpha_i^{-1}\alpha_j^{-1}\frac{1}{m}\sum_{r=1}^m \langle v_{\ell_1,r}, \mathcal{S}_i \rangle \langle v_{\ell_2,r}, \widetilde{\mathcal{S}}_j \rangle \widetilde{\exp}_{i,r}\widetilde{\exp}_{j,r}$$

$$B_3 := \alpha_i^{-1}\alpha_j^{-1}\frac{1}{m}\sum_{r=1}^m \langle v_{\ell_1,r}, \mathcal{S}_i \rangle \langle v_{\ell_2,r}, \widetilde{\mathcal{S}}_j \rangle \widetilde{\exp}_{i,r}\widetilde{\exp}_{j,r} - \alpha_i^{-1}\alpha_j^{-1}\frac{1}{m}\sum_{r=1}^m \langle v_{\ell_1,r}, \widetilde{\mathcal{S}}_i \rangle \langle v_{\ell_2,r}, \widetilde{\mathcal{S}}_j \rangle \widetilde{\exp}_{i,r}\widetilde{\exp}_{j,r}$$

$$B_4 := \alpha_i^{-1}\alpha_j^{-1}\frac{1}{m}\sum_{r=1}^m \langle v_{\ell_1,r}, \widetilde{\mathcal{S}}_i \rangle \langle v_{\ell_2,r}, \widetilde{\mathcal{S}}_j \rangle \widetilde{\exp}_{i,r}\widetilde{\exp}_{j,r} - \alpha_i^{-1}\widetilde{\alpha}_j^{-1}\frac{1}{m}\sum_{r=1}^m \langle v_{\ell_1,r}, \widetilde{\mathcal{S}}_i \rangle \langle v_{\ell_2,r}, \widetilde{\mathcal{S}}_j \rangle \widetilde{\exp}_{i,r}\widetilde{\exp}_{j,r}$$

$$B_5 := \alpha_i^{-1}\widetilde{\alpha}_j^{-1}\frac{1}{m}\sum_{r=1}^m \langle v_{\ell_1,r}, \widetilde{\mathcal{S}}_i \rangle \langle v_{\ell_2,r}, \widetilde{\mathcal{S}}_j \rangle \widetilde{\exp}_{i,r}\widetilde{\exp}_{j,r} - \widetilde{\alpha}_i^{-1}\widetilde{\alpha}_j^{-1}\frac{1}{m}\sum_{r=1}^m \langle v_{\ell_1,r}, \widetilde{\mathcal{S}}_i \rangle \langle v_{\ell_2,r}, \widetilde{\mathcal{S}}_j \rangle \widetilde{\exp}_{i,r}\widetilde{\exp}_{j,r}$$

Thus, we have
$$|[H_{\ell_1,\ell_2}]_{i,j}(W) - [H_{\ell_1,\ell_2}]_{i,j}(\widetilde{W})|/m^2 \leq |B_1| + |B_2| + |B_3| + |B_4| + |B_5|.$$

**To bound $B_1$**

We rewrite $B_1$ as

$$B_1 = \alpha_i^{-1}\alpha_j^{-1}\frac{1}{m}\sum_{r=1}^m \langle v_{\ell_1,r}, \mathcal{S}_i \rangle \langle v_{\ell_2,r}, \mathcal{S}_j \rangle (\exp(w_r^\top(x_i + x_j)) - \exp(\widetilde{w}_r^\top(x_i + x_j))).$$

Recall that $\|v_{\ell_1,r}\|_\infty \leq 2$ and $\|\mathcal{S}_i\|_1 \leq 1$. Thus, $|\langle v_{\ell_1,r}, \mathcal{S}_i \rangle| \leq 2$.

By Fact A.4, we know that $|\langle v_{\ell_1,r}, \mathcal{S}_i \rangle \langle v_{\ell_2,r}, \mathcal{S}_j \rangle| \leq 2 \cdot 2 = 4$. By Part 4 of Lemma B.1, with probability $1 - \delta/\operatorname{poly}(nd)$, we know that $|\alpha_i^{-1}| \leq \frac{1}{m}\exp(B)$.

We will condition on the above event is holding in the rest of the proof.

By Part 7 of Lemma B.1,

$$|\exp(\widetilde{w}_r^\top(x_i + x_j)) - \exp(w_r^\top(x_i + x_j))| \le 2R\exp(2B + 2R).$$

Finally, we know that

$$|B_1| \le \frac{8R}{m^2}\exp(5B).$$

**To bound $B_2$ and $B_3$**

We can rewrite $B_2$ as follows

$$|B_2| = |\alpha_i^{-1}\alpha_j^{-1}\frac{1}{m}\sum_{r=1}^m \langle v_{\ell_1,r}, \mathcal{S}_i\rangle\widetilde{\exp}_{i,r}\widetilde{\exp}_{j,r}(\langle v_{\ell_2,r}, \mathcal{S}_j\rangle - \langle v_{\ell_2,r}, \widetilde{\mathcal{S}}_j\rangle)|$$

$$\le \alpha_i^{-1}\alpha_j^{-1}\frac{1}{m}\sum_{r=1}^m |\langle v_{\ell_1,r}, \mathcal{S}_i\rangle|\widetilde{\exp}_{i,r}\widetilde{\exp}_{j,r}|(\langle v_{\ell_2,r}, \mathcal{S}_j\rangle - \langle v_{\ell_2,r}, \widetilde{\mathcal{S}}_j\rangle)|.$$

Following the similar strategy as $B_1$, by Part 13 of Lemma B.1, we know that

$$|B_2| \le \frac{1}{m}\exp(B) \cdot \frac{1}{m}\exp(B) \cdot 2 \cdot \exp(B + R) \cdot \exp(B + R) \cdot 4R\exp(4B + 3R)$$

$$\le \frac{8R}{m^2}\exp(9B).$$

Similarly, we have

$$|B_3| \le \frac{8R}{m^2}\exp(9B).$$

**To bound $B_4$ and $B_5$**

For the term $B_4$, we can rewrite

$$|B_4| = |(\alpha_j^{-1} - \widetilde{\alpha}_j^{-1}) \cdot \alpha_i^{-1}\frac{1}{m}\sum_{r=1}^m \langle v_{\ell_1,r}, \widetilde{\mathcal{S}}_i\rangle\langle v_{\ell_2,r}, \widetilde{\mathcal{S}}_j\rangle\widetilde{\exp}_{i,r}\widetilde{\exp}_{j,r}|$$

$$\le |\alpha_j^{-1} - \widetilde{\alpha}_j^{-1}| \cdot \alpha_i^{-1}\frac{1}{m}\sum_{r=1}^m |\langle v_{\ell_1,r}, \widetilde{\mathcal{S}}_i\rangle\langle v_{\ell_2,r}, \widetilde{\mathcal{S}}_j\rangle|\widetilde{\exp}_{i,r}\widetilde{\exp}_{j,r}.$$

Thus, by Part 9 of Lemma B.1, using similar proof strategy as $B_1$ as know

$$|B_4| \le \frac{R}{m}\exp(3B + 2R) \cdot \frac{1}{m}\exp(B) \cdot 2 \cdot 2 \cdot \exp(B + R) \cdot \exp(B + R)$$

$$\le \frac{4R}{m^2}\exp(7B).$$

Similarly, we have

$$|B_5| \le \frac{4R}{m^2}\exp(7B).$$

$\square$

## C   INDUCTION

In Section C.1, we provide the proof of our main result. In Section C.2, we provide an induction lemma for weights part. In Section C.3, we provide an induction lemma for loss part. In Section C.4, we provide an induction lemma for gradient part.

## C.1 MAIN RESULT

Our main result is presented as follows.

**Theorem C.1** (Main result. Restatement of Theorem 4.2). *For any $\epsilon, \delta \in (0, 0.1)$, if the following conditions hold*

- *Let $\lambda = \lambda_{\min}(H^*) > 0$*

- *Let $m = \Omega(\lambda^{-2} n^2 d^2 \exp(18B) \log^2(nd/\delta))$*

- *Let $\eta = 0.1\lambda/(mn^2 d^2 \exp(16B))$*

- *Let $\widehat{T} = \Omega((m\eta\lambda)^{-1} \log(nd/\epsilon)) = \Omega(\lambda^{-2} n^2 d^2 \exp(16B) \cdot \log(nd/\epsilon))$*

*Then, after $\widehat{T}$ iterations, with probability at least $1 - \delta$, we have*

$$\|F(\widehat{T}) - Y\|_F^2 \leq \epsilon.$$

*Proof of Theorem 4.2.* Let $\sigma = 1$. We have $\|F(0) - Y\|_F^2 \leq nd$ by Lemma D.3.

Using the choice of $\widehat{T}$, it follows directly from the alternative application of Lemma C.3 and Lemma C.2.

Since $\exp(\Theta(B)) = (nd)^{o(1)}$, we can simplify the $nd\exp(\Theta(B)) = (nd)^{1+o(1)}$. □

## C.2 INDUCTION PART 1. FOR WEIGHTS

We provide an induction lemma for weights part.

**Lemma C.2** (Induction Part 1. For Weights). *Let $\tau$ be a fixed integer.*

*If the below conditions are true*

- *General Property 1. Let $\lambda = \lambda_{\min}(H^*) > 0$*

- *General Property 2. $\eta = 0.1\lambda/(mn^2 d^2 \exp(16B))$*

- *General Property 3. Let $D$ be defined as Definition A.2*

- *General Property 4. $D < R = \lambda/(2nd\exp(10B))$*

- *General Property 5. $m = \Omega(\lambda^{-2} n^2 d^2 \exp(18B) \log^2(nd/\delta))$*

- **Weights Property.** $\|w_r(i) - w_r(0)\|_2 \leq R$ *for all $i \in [\tau]$*

- **Loss Property.** $\|F(i) - Y\|_F^2 \leq \|F(0) - Y\|_F^2 \cdot (1 - m\eta\lambda/2)^i, \forall i \in [\tau]$

- **Gradient Property.** $\eta\|\Delta w_r(i)\|_2 \leq 0.01, \forall r \in [m], \forall i \in [\tau]$

*Then, for $\tau + 1$ and $\forall r \in [m]$, we have*

$$\|w_r(\tau + 1) - w_r(0)\|_2 \leq D.$$

*Proof.* We have

$$\eta \sum_{i=0}^{\infty} (1 - m\eta\lambda/2)^{i/2}$$

$$\leq \eta \sum_{i=0}^{\infty} (1 - m\eta\lambda/4)^i$$

$$\leq \eta \frac{1}{m\eta\lambda/4}$$

$$\leq \frac{4}{m\lambda} \tag{9}$$

where the first step is due to the Fact A.5, the second stepis due to the Fact A.7, the last step is because of simple algebra.

We use the gradient's norm to measure the weights difference:

$$\|w_r(0) - w_r(\tau + 1)\|_2$$

$$\leq \eta \sum_{i=0}^{\tau} \|\Delta w_r(i)\|_2$$

$$\leq \eta \sum_{i=0}^{\tau} \exp(3B)\sqrt{nd} \cdot \|F(i) - Y\|_F$$

$$\leq \eta \exp(3B)\sqrt{nd} \sum_{i=0}^{\tau} (1 - m\eta\lambda/2)^{i/2} \cdot \|F(0) - Y\|_F$$

$$\leq 4m^{-1}\lambda^{-1} \exp(3B)\sqrt{nd} \cdot \|F(0) - Y\|_F$$

$$= D$$

where the first step follows from $w_r(i+1) - w_r(i) = \eta \cdot \Delta w_r(i)$, the second step follows from Lemma D.1 for $\tau$ times, the third step follows from **Loss Property** in Lemma statement, the fourth step follows from Eq. (9), the last step is from General Property 3 in Lemma statement. $\qquad \square$

## C.3 Induction Part 2. For Loss

We provide an induction lemma for loss part.

**Lemma C.3** (Induction Part 2. For Loss). *Let $\tau$ be a fixed integer.*

*If the following conditions hold*

- *General Property 1. Let $\lambda = \lambda_{\min}(H^*) > 0$*

- *General Property 2. $\eta = 0.1\lambda/(mn^2d^2\exp(16B))$*

- *General Property 3. Let $D$ be defined as Definition A.2*

- *General Property 4. $D < R = \lambda/(2nd\exp(10B))$*

- *General Property 5. $m = \Omega(\lambda^{-2}n^2d^2\exp(18B)\log^2(nd/\delta))$*

- **Weights Property.** $\|w_r(\tau) - w_r(0)\|_2 \leq D < R, \forall r \in [m]$

- **Loss Property.** $\|F(i) - Y\|_F^2 \leq \|F(0) - Y\|_F^2 \cdot (1 - m\eta\lambda/2)^i, \forall i \in [\tau]$

- **Gradient Property.** $\eta\|\Delta w_r(i)\|_2 \leq 0.01 \ \forall r \in [m], \forall i \in [\tau]$

*Then we have*

$$\|F(\tau + 1) - Y\|_F^2 \leq (1 - m\eta\lambda/4)^{\tau+1} \cdot \|F(0) - Y\|_F^2.$$

*Proof.* We have

$$\|F(\tau) - Y\|_F^2 \leq \|F(\tau - 1) - Y\|_F^2 \cdot (1 - m\eta\lambda/2)$$

which follows Lemma E.2.

Thus, we complete the proof by induction.

$\qquad \square$

## C.4   INDUCTION PART 3. FOR GRADIENT

We provide an induction lemma for gradient part.

**Lemma C.4** (Induction Part 3. For Gradient). *Let $\tau$ be a fixed integer.*

*If the following conditions hold*

- *General Property 1. Let $\lambda = \lambda_{\min}(H^*) > 0$*

- *General Property 2. $\eta = 0.1\lambda/(mn^2d^2 \exp(16B))$*

- *General Property 3. Let $D$ be defined as Definition A.2*

- *General Property 4. $D < R = \lambda/(2nd \exp(10B))$*

- *General Property 5. $m = \Omega(\lambda^{-2}n^2d^2 \exp(18B) \log^2(nd/\delta))$*

- **Weights Property.** $\|w_r(\tau) - w_r(0)\|_2 \leq D < R, \forall r \in [m]$

- **Loss Property.** $\|F(i) - Y\|_F^2 \leq \|F(0) - Y\|_F^2 \cdot (1 - m\eta\lambda/2)^i, \forall i \in [\tau]$

- **Gradient Property.** $\eta\|\Delta w_r(i)\|_2 \leq 0.01 \ \forall r \in [m], \forall i \in [\tau]$

*Then we have*

$$\eta\|\Delta w_r(\tau + 1)\|_2 \leq 0.01, \forall r \in [m]$$

*Proof.* This is trivially follows from Lemma D.1 and Lemma D.2.

□

## D   INDUCTION PART 1: FOR WEIGHTS

In Section D.1, we propose the lemma for bounding gradient and its corresponding proof. In Section D.2, we propose the bounding initialization loss and its corresponding proof.

### D.1   BOUNDING THE GRADIENT AT ANY TIME

In this section, we bound the gradient.

**Lemma D.1.** *If the following condition hold,*

- *Let $B > 1$ denote a parameter be defined as Definition 4.1*

- *Let $R \in (0, 0.01)$*

- *$\|w_r(\tau) - w_r(0)\|_2 \leq R$*

- *Let $v_{\ell,r} = a_{\ell,r} \cdot \mathbf{1}_m - a_\ell \in \mathbb{R}^m$, for any $\ell \in [d]$ and for any $r \in [m]$*

*For any timestamp $\tau$, we have*

$$\|\Delta w_r(\tau)\|_2 \leq \exp(3B)\sqrt{nd} \cdot \|F(\tau) - Y\|_F.$$

*Proof.* We have

$$\|\Delta w_r(\tau)\|_2 = \left\| m \sum_{i=1}^n \sum_{\ell=1}^d (y_{\ell,i} - F_{\ell,i}) \cdot x_i \cdot \langle v_{\ell,r}, \mathcal{S}_i(\tau)\rangle \cdot \mathcal{S}_{i,r}(\tau) \right\|_2$$

$$\leq \exp(3B) \sum_{i=1}^n \sum_{\ell=1}^d |y_{\ell,i} - F_{\ell,i}(\tau)|$$

$$\leq \exp(3B)\sqrt{nd} \cdot \|F(\tau) - Y\|_F$$

where the first step follows from Claim 3.4 and Definition 3.3, the second step follows from $|\langle v_{\ell,r}, \mathcal{S}_i \rangle| \leq 2$ and $|\mathcal{S}_{i,r}| \leq \exp(2B + 2R)/m$ by Part 11 of Lemma B.1, the last step follows from Cauchy-Schwartz inequality.

$\square$

**Lemma D.2.** *If the following conditions hold,*

- $\eta = 0.1\lambda/(mn^2d^2\exp(16B))$

- $\|w_r(\tau) - w_r(0)\|_2 \leq R$

*Then, for any timestamp $\tau$, we have*

$$\eta\|\Delta w_r(\tau)\|_2 \leq 0.01$$

*Proof.* This trivially follows from Lemma D.1 and choice of $\eta$.

$\square$

## D.2    BOUNDING THE INITIALIZATION LOSS

In this section, we bound the initialization loss.

**Lemma D.3.** *We have*

$$\|F(0) - Y\|_F \leq O(\sqrt{nd}).$$

*Proof.* This trivially follows from $\|y_i\| \leq 1, \forall i \in [n]$ and symmetric initialization from Definition 3.7.

$\square$

## E    INDUCTION PART 2: FOR LOSS

In Section E.1, we decompose the loss $\|F(k+1) - Y\|_F^2$ into four parts, namely $C_0, C_1, C_2$, and $C_3$. In Section E.2, we show our choices of $m$ and $\eta$. In Section E.3, we establish bounds for $C_0$. In Section E.4, we establish bounds for $C_1$. In Section E.5, we establish bounds for $C_2$. In Section E.6, we establish bounds for $C_3$.

### E.1    DECOMPOSITION FOR $\|\operatorname{vec}(F(\tau+1) - Y)\|_2^2$

Here, we decompose the loss $\|\operatorname{vec}(F(\tau+1) - Y)\|_2^2$ into four parts $C_0, C_1, C_2$ and $C_3$.

**Lemma E.1.** *Assuming the following condition is met:*

- *Let $\lambda = \lambda_{\min}(H^*)$*

- *Let $\alpha_i(\tau) := \langle \exp(W(\tau)^\top x_i), \mathbf{1}_m \rangle$.*

- *Let scalar $v_{0,\ell,i} \in \mathbb{R}$ be defined as follows*

$$v_{0,\ell,i} := m \sum_{r \in [m]} a_{\ell,r}(\alpha_i(\tau+1)^{-1} - \alpha_i(\tau)^{-1}) \cdot (\exp(\langle w_r(\tau+1), x_i \rangle))$$

- *Let scalar $v_{1,\ell,i} \in \mathbb{R}$ be defined as follows*

$$v_{1,\ell,i} := m \sum_{r=1}^{m} a_{\ell,r} \cdot \alpha_i(\tau)^{-1} \exp((\langle w_r(\tau), x_i \rangle) \cdot (-\eta \langle \Delta w_r(\tau), x_i \rangle)$$

- *Let scalar $v_{2,\ell,i} \in \mathbb{R}$ be defined as follows*

$$v_{2,\ell,i} := m \sum_{r=1}^{m} a_{\ell,r} \cdot \alpha_i(\tau)^{-1} \exp((\langle w_r(\tau), x_i \rangle) \cdot \eta^2 \cdot \Theta(1) \cdot \langle \Delta w_r(\tau), x_i \rangle^2$$

- **Gradient Property.** $\eta\|\Delta w_r(i)\|_2 \le 0.01, \forall r \in [m], \forall i \in [\tau]$

- $C_0 = 2\langle \text{vec}(F(\tau) - Y), \text{vec}(v_0)\rangle$

- $C_1 = 2\langle \text{vec}(F(\tau) - Y), \text{vec}(v_1)\rangle$

- $C_2 = 2\langle \text{vec}(F(\tau) - Y), \text{vec}(v_2)\rangle$

- $C_3 = \|F(\tau + 1) - F(\tau)\|_F^2$

*then*

$$\|F(\tau + 1) - Y\|_F^2 = \|F(t) - Y\|_F^2 + C_0 + C_1 + C_2 + C_3.$$

*Proof.* The expression $\|Y - F(\tau+1)\|_F^2 = \|\text{vec}(Y - F(\tau+1))\|_2^2$ can be rewritten in the following:

$$\| \text{vec}(Y - F(\tau + 1))\|_2^2$$
$$= \| \text{vec}(Y - F(\tau) - (F(\tau + 1) - F(\tau)))\|_2^2$$
$$= \| \text{vec}(Y - F(\tau))\|_2^2 - 2\text{vec}(Y - F(\tau))^\top \text{vec}(F(\tau + 1) - F(\tau)) + \| \text{vec}(F(\tau + 1) - F(\tau))\|_2^2. \tag{10}$$

where the first step follows from simple algebra, the last step follows from Fact A.3.

Recall the update rule (Definition 3.5),

$$w_r(\tau + 1) = w_r(\tau) - \eta \cdot \Delta w_r(\tau)$$

In the following manner, $\forall \ell \in [d]$, we can express $F_\ell(\tau + 1) - F_\ell(\tau) \in \mathbb{R}^n$:

$$F_{\ell,i}(\tau + 1) - F_{\ell,i}(\tau)$$
$$= m \sum_{r\in[m]} a_{\ell,r} \cdot (\alpha_i(\tau + 1)^{-1} \exp(\langle w_r(\tau + 1), x_i\rangle) - \alpha_i(\tau)^{-1} \exp(\langle w_r(\tau), x_i\rangle))$$
$$= + m \sum_{r\in[m]} a_{\ell,r}(\alpha_i(\tau + 1)^{-1} - \alpha_i(\tau)^{-1}) \cdot (\exp(\langle w_r(\tau + 1), x_i\rangle))$$
$$+ m \sum_{r\in[m]} a_{\ell,r}\alpha_i(\tau)^{-1} \cdot (\exp(\langle w_r(\tau + 1), x_i\rangle) - \exp(\langle w_r(\tau), x_i\rangle))$$
$$= + m \sum_{r\in[m]} a_{\ell,r}(\alpha_i(\tau + 1)^{-1} - \alpha_i(\tau)^{-1}) \cdot (\exp(\langle w_r(\tau + 1), x_i\rangle))$$
$$+ m \sum_{r\in[m]} a_{\ell,r} \cdot \alpha_i(\tau)^{-1} \exp((\langle w_r(\tau), x_i\rangle) \cdot (\exp(-\eta\langle\Delta w_r(\tau), x_i\rangle) - 1)$$
$$= + m \sum_{r\in[m]} a_{\ell,r}(\alpha_i(\tau + 1)^{-1} - \alpha_i(\tau)^{-1}) \cdot (\exp(\langle w_r(\tau + 1), x_i\rangle))$$
$$+ m \sum_{r\in[m]} a_{\ell,r} \cdot \alpha_i(\tau)^{-1} \exp((w_r(\tau)^\top x_i) \cdot (-\eta\langle\Delta w_r(\tau), x_i\rangle + \Theta(1)\eta^2\langle\Delta w_r(\tau), x_i\rangle^2)$$
$$= v_{0,\ell,i} + v_{1,\ell,i} + v_{2,\ell,i}$$

where the first step is due to the definition of $F_{\ell,i}(\tau)$, the second step is from the simple algebra, the third step is due to $|\eta\Delta w_r(\tau)^\top x_i| \le 0.01$ (due to **Gradient Property** and $\|x_i\|_2 \le 1$), the fourth step follows from the Fact A.8, the last step follows from

$$v_{0,\ell,i} := m \sum_{r\in[m]} a_{\ell,r}(\alpha_i(\tau + 1)^{-1} - \alpha_i(\tau)^{-1}) \cdot (\exp(\langle w_r(\tau + 1), x_i\rangle))$$

$$v_{1,\ell,i} := m \sum_{r=1}^m a_{\ell,r} \cdot \alpha_i(\tau)^{-1} \exp((\langle w_r(\tau), x_i\rangle) \cdot (-\eta\langle\Delta w_r(\tau), x_i\rangle)$$

$$v_{2,\ell,i} := m \sum_{r=1}^{m} a_{\ell,r} \cdot \alpha_i(\tau)^{-1} \exp((\langle w_r(\tau), x_i \rangle) \cdot \eta^2 \cdot \Theta(1) \cdot \langle \Delta w_r(\tau), x_i \rangle^2$$

Here $v_{0,\ell,i}$ and $v_{1,\ell,i}$ are linear in $\eta$ and $v_{2,\ell,i}$ is quadratic in $\eta$. Thus, $v_{0,\ell,i}$ and $v_{1,\ell,i}$ are the first order term, and $v_{2,\ell,i}$ is the second order term.

We can rewrite the second term in the Eq. (10) above as below:

$$\begin{aligned}
&\langle \mathrm{vec}(Y - F(\tau)), \mathrm{vec}(F(\tau + 1) - F(\tau)) \rangle \\
&= \langle \mathrm{vec}(Y - F(\tau)), \mathrm{vec}(v_0 + v_1 + v_2) \rangle \\
&= \langle \mathrm{vec}(Y - F(\tau)), \mathrm{vec}(v_0) \rangle + \langle \mathrm{vec}(Y - F(\tau)), \mathrm{vec}(v_1) \rangle + \langle \mathrm{vec}(Y - F(\tau)), \mathrm{vec}(v_2) \rangle
\end{aligned}$$

Therefore, we can conclude that

$$\|F(\tau + 1) - Y\|_F^2 = \|F(\tau) - Y\|_F^2 + C_0 + C_1 + C_2 + C_3.$$

$\square$

### E.2 Choice of Parameters

Here, we show our choice of parameters $m, \eta, R, B$.

**Lemma E.2.** *If the below conditions are true*

- *Condition 1. Let $\lambda = \lambda_{\min}(H^*) > 0$*

- *Condition 2. $m = \Omega(\lambda^{-2} n^2 d^2 \exp(18B) \log^2(nd/\delta))$*

- *Condition 3. $\eta = 0.1\lambda/(mn^2 d^2 \exp(16B))$*

- *Condition 4. $R = \lambda/(2nd \exp(10B))$*

  - *Required by Claim E.5*

- *Condition 5. $B = \max\{C\sigma\sqrt{\log(nd/\delta)}, 1\}$*

- *Condition 6. $D = 4m^{-1}\lambda^{-1}\exp(3B)\sqrt{nd} \cdot \|F(0) - Y\|_F$*

- *Condition 7. $D < R$*

- *Condition 8. $\eta\|\Delta w_r(\tau)\|_2 \leq 0.01, \forall r \in [m]$*

  - *Required by Lemma E.1, Claim E.3 and Claim E.7*

*Then it holds that*

$$\|F(\tau + 1) - Y\|_F^2 \leq \|F(\tau) - Y\|_F^2 \cdot (1 - m\eta\lambda/2)$$

*holds with probability at least $1 - \delta$.*

*Proof.* We can show

$$\begin{aligned}
&\|F(\tau + 1) - Y\|_F^2 \\
&= \|F(\tau) - Y\|_F^2 + C_0 + C_1 + C_2 + C_3 \\
&\leq (1 - 0.8m\eta\lambda + 0.1m\eta\lambda + 2m\eta^2 n^2 d^2 \exp(9B) + \eta^2 m^2 \cdot n^2 d^2 \cdot \exp(16B)) \cdot \|F(\tau) - Y\|_F^2 \\
&\leq (1 - 0.7m\eta\lambda + 2\eta^2 m^2 \cdot n^2 d^2 \cdot \exp(16B)) \cdot \|F(\tau) - Y\|_F^2.
\end{aligned}$$

where the first step follows from Lemma E.1, the second step follows from Lemma E.3 for $C_0$, Lemma E.4, Claim E.5 for $C_1$, Claim E.6 for $C_2$ and Claim E.7 for $C_3$, the last step follows from the simple algebra.

**Choice of $\eta$.** Next, we want to choose $\eta$ such that

$$(1 - 0.7m\eta\lambda + 2\eta^2 m^2 \cdot n^2 d^2 \cdot \exp(16B)) \leq (1 - m\eta\lambda/2). \tag{11}$$

Using the choice of $\eta$ in Condition 3

$$2\eta^2 m^2 \cdot n^2 d^2 \cdot \exp(16B) \leq 0.2m\eta\lambda$$

This indicates:

$$\|F(\tau + 1) - Y\|_F^2 \leq (1 - m\eta\lambda/2) \cdot \|F(\tau) - Y\|_F^2. \tag{12}$$

**Lower bound for $m$, over-parametrization size.** We require the following conditions

- $m \geq \Omega(\lambda^{-2}n^2 d \exp(18B) \log^2(nd/\delta))$ (required by Lemma E.3)

- $m \geq \Omega(\lambda^{-2}n^2 d \exp(12B) \log^2(nd/\delta))$ (required by Lemma E.4)

- $D = 4m^{-1}\lambda^{-1} \exp(3B)\sqrt{nd} \cdot \|F(0) - Y\|_F < R = \lambda/(2nd\exp(10B))\}$ (required by Condition 7.)

Therefore, by $\|Y - F(0)\|_F = O(\sqrt{nd})$ from Lemma D.3, it suffices to choose:

$$m = \Omega(\lambda^{-2}n^2 d^2 \exp(18B) \log^2(nd/\delta)).$$

$\square$

### E.3 Bounding $C_0$

Here, we explain about how to bound $C_0$.

**Lemma E.3.** *If the following conditions hold*

- *Let scalar $v_{0,\ell,i} \in \mathbb{R}$ be defined as follows*

$$v_{0,\ell,i} := m \sum_{r \in [m]} a_{\ell,r}(\alpha_i(\tau + 1)^{-1} - \alpha_i(\tau)^{-1}) \cdot (\exp(\langle w_r(\tau + 1), x_i\rangle))$$

- *Let $\alpha_i(\tau) := \langle \exp(W(\tau)^\top x_i), \mathbf{1}_m\rangle$.*

- *Let $m \geq \Omega(\lambda^{-2}n^2 d \exp(18B) \log^2(nd/\delta))$*

- **Gradient Property.** *$\eta\|\Delta w_r(i)\|_2 \leq 0.01, \forall r \in [m], \forall i \in [\tau]$*

- *We define $C_0$ as follows*

$$C_0 = 2\langle \text{vec}(F(\tau) - Y), \text{vec}(v_0)\rangle$$

*Here $\text{vec}(v_0) \in \mathbb{R}^{nd}$ is the vectorization of $v_0 \in \mathbb{R}^{n \times d}$ and $\text{vec}(F(\tau) - Y) \in \mathbb{R}^{nd}$ is the vectorization of $F(\tau) - Y \in \mathbb{R}^{n \times d}$.*

*Then we have*

$$|C_0| \leq 0.1m\eta\lambda \cdot \|F(\tau) - Y\|_F^2$$

*Proof.* We can rewrite $v_{0,\ell,i}$ as follows:

$$v_{0,\ell,i} = m \sum_{r=1}^{m} a_{\ell,r}((\alpha_i(\tau + 1))^{-1} - \alpha_i(\tau)^{-1}) \exp(\langle w_r(\tau + 1), x_i\rangle)$$

$$= m \sum_{r=1}^{m} a_{\ell,r}\alpha_i(\tau + 1)^{-1}\alpha_i(\tau)^{-1} \cdot (\langle \mathbf{1}_m, \exp(W(\tau + 1)x_i) - \exp(W(\tau)x_i)\rangle) \exp(\langle w_r(\tau + 1), x_i\rangle)$$

$$= m \sum_{r=1}^{m} a_{\ell,r} \alpha_i(\tau+1)^{-1} \alpha_i(\tau)^{-1} \Big( \sum_{r_2=1}^{m} \exp(w_{r_2}(\tau+1)x_i) - \exp(w_{r_2}(\tau)x_i) \Big) \exp(\langle w_r(\tau+1), x_i \rangle)$$

$$= m \Big( \sum_{r=1}^{m} a_{\ell,r} \alpha_i(\tau+1)^{-1} \alpha_i(\tau)^{-1} \sum_{r_2=1}^{m} -\eta \langle \Delta w_{r_2}(\tau), x_i \rangle \exp(w_{r_2}(\tau)x_i) \exp(\langle w_r(\tau+1), x_i \rangle) + \eta^2 \Delta_2 \Big)$$

$$= m \Big( \underbrace{\sum_{r=1}^{m} a_{\ell,r} \sum_{r_2=1}^{m} -\eta \langle \Delta w_{r_2}(\tau), x_i \rangle \mathcal{S}_{i,r_2}(\tau) \cdot \mathcal{S}_{i,r}(\tau+1)}_{\text{first order term}} + \underbrace{\eta^2 \Delta_2}_{\text{second order term}} \Big) \tag{13}$$

where the first step follows from lemma statement, the second step follows from $a^{-1} - b^{-1} = \frac{b-a}{ab}$, the third step follows from simple algebra, the fourth step follows from simple algebra, and the last step follows from $|\eta \Delta w_r(\tau)^\top x_i| \le 0.01$ (due to **Gradient Property** and $\|x_i\|_2 \le 1$).

The second order term $\eta^2 \Delta_2$ in Eq. (13) can be bounded in a similar way as the proof of Claim E.6.

Further, we can rewrite the first-order term in Eq. (13)

$$m \sum_{r=1}^{m} a_{\ell,r} \sum_{r_2=1}^{m} -\eta \langle \Delta w_{r_2}(\tau), x_i \rangle \mathcal{S}_{i,r_2}(\tau) \cdot \mathcal{S}_{i,r}(\tau+1) = m^2 (Q_{1,i,\ell} + Q_{2,i,\ell}) \tag{14}$$

where

$$Q_{1,i,\ell} := \frac{1}{m} \sum_{r=1}^{m} a_{\ell,r} (-\eta \langle \Delta w_r(\tau), x_i \rangle) \mathcal{S}_{i,r}(\tau) \cdot \mathcal{S}_{i,r}(\tau+1)$$

$$Q_{2,i,\ell} := \frac{1}{m} \sum_{r=1}^{m} a_{\ell,r} \sum_{r_2 \ne r} (-\eta \langle \Delta w_{r_2}(\tau), x_i \rangle) \mathcal{S}_{i,r_2}(\tau) \cdot \mathcal{S}_{i,r}(\tau+1)$$

Let us consider how to handle the first term in Eq. (13),

$$Q_{1,i,\ell} = \frac{1}{m} \sum_{r=1}^{m} a_{\ell,r} (-\eta \langle \Delta w_r(\tau), x_i \rangle) \mathcal{S}_{i,r}(\tau) \cdot \mathcal{S}_{i,r}(\tau+1)$$

$$= \sum_{r=1}^{m} a_{\ell,r} \mathcal{S}_{i,r} \cdot \mathcal{S}_{i,r}(\tau+1) \Big( -\eta \sum_{j=1}^{n} \sum_{\ell_2=1}^{d} (F_{\ell_2,j}(\tau) - y_{\ell_2,j}) \cdot \Big( (\langle a_{\ell_2,r} \cdot \mathbf{1}_m - a_{\ell_2}, \mathcal{S}_j \rangle) \cdot \mathcal{S}_{j,r} \Big) \cdot x_j^\top \Big) x_i$$

where the second step follows from computing $\Delta w_r(\tau)$ explicitly (see Claim 3.4).

Similarly as proof of Lemma E.4, we can use concentration to bound

$$\sum_{i=1}^{n} \sum_{\ell=1}^{d} Q_{1,i,\ell} (F_{\ell,i} - y_{\ell,i})$$

Note that $0 < \mathcal{S}_{j,r} < \frac{\exp(3B)}{m}$ by Part 11 of Lemma B.1. The above small term is equivalent to

$$-\eta \frac{\exp(9B)}{m^3} \cdot \sum_{i=1}^{n} \sum_{j=1}^{n} \sum_{r=1}^{m} \sum_{\ell=1}^{d} \sum_{\ell_2=1}^{d} (F_{\ell_2,j}(\tau) - y_{\ell_2,j}) \cdot \sigma_{i,j,r,\ell,\ell_2} \cdot C_{i,j,r,\ell,\ell_2} \cdot (F_{\ell,i}(\tau) - y_{\ell,i}),$$

where $\sigma_{i,\ell,\ell_2,j,r} \sim [-1,+1]$ and $|C_{i,\ell,\ell_2,j,r}| \le 10$. We define

$$P_{1,r,\ell,\ell_2} := (F_{\ell_2,j} - y_{\ell_2,j}) \sigma_{i,j,r,\ell,\ell_2} C_{i,j,r,\ell,\ell_2} (F_{\ell,i} - y_{\ell,i})$$

Similarly as Lemma E.4, for each fixed $i, j \in [n]$, using Hanson-Wright inequality (Lemma A.10), we can show

$$\Pr[ |\sum_{r=1}^{m} \sum_{\ell=1}^{d} \sum_{\ell_2=1}^{d} P_{1,r,\ell,\ell_2}| \le 100 \|F_j - y_j\|_2 \|F_i - y_i\|_2 \cdot \sqrt{md} \log(nd/\delta) ]$$

$$\geq 1 - \delta/\operatorname{poly}(nd).$$

By mean inequality, we have

$$\sum_{i=1}^{n}\sum_{j=1}^{n}\|F_j - y_j\|_2 \cdot \|F_i - y_i\|_2 \leq n\|F - y\|_F^2.$$

Thus, we have the first term with probability at least $1 - \operatorname{poly}(nd)$, such that

$$|\sum_{i=1}^{n}\sum_{\ell=1}^{d}Q_{1,i,\ell}(F_{\ell,i} - y_{\ell,i})| \leq \eta\frac{n\exp(9B)}{m^3}\|F - y\|_F^2\sqrt{md}\log(nd/\delta)$$

Similarly, we can compute

$$\sum_{i=1}^{n}\sum_{\ell=1}^{d}Q_{2,i,\ell}(F_{\ell,i} - y_{\ell,i})$$

Using Hanson-Wright inequality (Lemma A.10), we have the second term with probability at least $1 - \operatorname{poly}(nd)$, such that

$$|\sum_{i=1}^{n}\sum_{\ell=1}^{d}Q_{2,i,\ell}(F_{\ell,i} - y_{\ell,i})| \leq \eta\frac{n\exp(9B)}{m^2}\|F - y\|_F^2\sqrt{md}\log(nd/\delta)$$

Thus, we can complete the proof by the Lemma statement $m \geq \Omega(\lambda^{-2}n^2 d\exp(18B)\log^2(nd/\delta))$. $\qquad\square$

### E.4 BOUNDING $C_1$

Here, we give the bound of the first order term $C_1$. Note that this term is making progress.

**Lemma E.4.** *Assuming the following condition is met:*

- *Let $\lambda = \lambda_{\min}(H^*)$*

- *Let $\alpha_i(\tau) := \langle\exp(W(\tau)^\top x_i), \mathbf{1}_m\rangle$*

- *Let $m \geq \Omega(\lambda^{-2}n^2 d\exp(12B)\log^2(nd/\delta))$*

- *Let scalar $v_{1,\ell,i} \in \mathbb{R}$ be defined as follows*

$$v_{1,\ell,i} := m\sum_{r=1}^{m}a_{\ell,r}\cdot\alpha_i(\tau)^{-1}\exp((\langle w_r(\tau), x_i\rangle)\cdot(-\eta\langle\Delta w_r(\tau), x_i\rangle)$$

- *$C_1 = 2\langle\operatorname{vec}(F(\tau) - Y), \operatorname{vec}(v_1)\rangle$*

*then*

$$C_1 \leq -1.6m\eta\operatorname{vec}(F(\tau) - Y)^\top H(\tau)\operatorname{vec}(F(\tau) - Y).$$

*Proof.* To simplify the notation, we omit writing $(\tau)$ in $\mathcal{S}_{i,r}(\tau)$. Then, we can express $v_{1,\ell,i} \in \mathbb{R}$ as follows:

$$v_{1,\ell,i} = m\sum_{r\in[m]}a_{\ell,r}\cdot\mathcal{S}_{i,r}\cdot(-\eta\langle x_i, \Delta w_r(\tau)\rangle)$$

$$= m^2\sum_{r\in[m]}a_{\ell,r}\cdot\mathcal{S}_{i,r}\cdot(-\eta\sum_{j=1}^{n}\sum_{\ell_2=1}^{d}(F_{\ell_2,j}(\tau) - y_{\ell_2,j})\cdot\left((\langle a_{\ell_2,r}\cdot\mathbf{1}_m - a_{\ell_2}, \mathcal{S}_j\rangle)\cdot\mathcal{S}_{j,r}\right)\cdot x_j^\top)x_i$$

$$= m^2(Q_{1,\ell,i} + Q_{2,\ell,i}) \tag{15}$$

where the second step using equation for $\Delta w_r(\tau)$ (see Claim 3.4).

Note that $\langle a_{\ell,r} \cdot \mathbf{1}_m, S_i \rangle = a_{\ell,r}$, so in the above equation,

$$Q_{1,\ell,i} := \sum_{r \in [m]} \langle a_{\ell,r} \cdot \mathbf{1}_m - a_\ell, S_i \rangle \cdot \mathcal{S}_{i,r} \cdot (-\eta \sum_{j=1}^{n} \sum_{\ell_2=1}^{d} (F_{\ell_2,j}(\tau) - y_{\ell_2,j}) \cdot \left( (\langle a_{\ell_2,r} \cdot \mathbf{1}_m - a_{\ell_2}, \mathcal{S}_j \rangle) \cdot \mathcal{S}_{j,r} \right) \cdot x_j^\top) x_i$$

$$Q_{2,\ell,i} := \sum_{r \in [m]} \langle a_\ell, S_i \rangle \cdot \mathcal{S}_{i,r} \cdot (-\eta \sum_{j=1}^{n} \sum_{\ell_2=1}^{d} (F_{\ell_2,j}(\tau) - y_{\ell_2,j}) \cdot \left( (\langle a_{\ell_2,r} \cdot \mathbf{1}_m - a_{\ell_2}, \mathcal{S}_j \rangle) \cdot \mathcal{S}_{j,r} \right) \cdot x_j^\top) x_i$$

The quantity $\sum_{i \in [n]} \sum_{\ell \in [d]} Q_{1,\ell,i}(F_{\ell,i} - Y_{\ell,i})$ is corresponding to first term $(Q_{1,\ell,i})$ in Eq. (15). It is

$$\sum_{i \in [n]} \sum_{\ell \in [d]} Q_{1,\ell,i}(F_{\ell,i} - Y_{\ell,i}) = -\frac{1}{m} \eta \operatorname{vec}(F(\tau) - Y)^\top H(\tau)^\top \operatorname{vec}(F(\tau) - Y) \qquad (16)$$

The quantity $\sum_{i \in [n]} \sum_{\ell \in [d]} Q_{2,\ell,i}(F_{\ell,i} - Y_{\ell,i})$ is corresponding to second term $(Q_{2,\ell,i})$ in Eq. (15). Note that $0 < \mathcal{S}_{j,r} < \frac{\exp(3B)}{m}$ by Part 11 of Lemma B.1. The quantity,

$$\sum_{i \in [n]} \sum_{\ell \in [d]} Q_{2,\ell,i}(F_{\ell,i} - Y_{\ell,i}) \qquad (17)$$

is equivalent to

$$-\eta \frac{\exp(6B)}{m^2} \cdot \sum_{i=1}^{n} \sum_{j=1}^{n} \sum_{r=1}^{m} \sum_{\ell=1}^{d} \sum_{\ell_2=1}^{d} (F_{\ell_2,j}(\tau) - y_{\ell_2,j}) \cdot \sigma_{i,j,r,\ell,\ell_2} \cdot C_{i,j,r,\ell,\ell_2} \cdot (F_{\ell,i}(\tau) - y_{\ell,i}),$$

where $\sigma_{i,j,r,\ell,\ell_2} \in \{-1, +1\}$ and $|C_{i,j,r,\ell,\ell_2}| \leq 10$.

Note that there are four cases

- $i = j, \ell = \ell_2$, this is a p.s.d. case that always makes progress, thus we can drop it.

- $i \neq j, \ell = \ell_2$ we will use random variable $P_1$ to handle

- $i = j, \ell \neq \ell_2$ we will use random variable $P_2$ to handle

- $i \neq j, \ell \neq \ell_2$ we will use random variable $P_2$ to handle

For each fixed $i, j \in [n]$. We define

$$P_{1,r,\ell} := (F_{\ell,j} - y_{\ell,j})\sigma_{i,j,r,\ell}C_{i,j,r,\ell}(F_{\ell,i} - y_{\ell,i})$$
$$P_{2,r,\ell,\ell_2} := (F_{\ell_2,j} - y_{\ell_2,j})\sigma_{i,j,r,\ell,\ell_2}C_{i,j,r,\ell,\ell_2}(F_{\ell,i} - y_{\ell,i})$$

The random variables related to $P_{1,r,\ell}$ are the following

$$\sum_{r=1}^{m} \sum_{\ell=1}^{d} P_{1,r,\ell}$$

The random variables related to $P_{2,r,\ell,\ell_2}$ are the following

$$\sum_{r=1}^{m} \sum_{\ell=1}^{d} \sum_{\ell_2=1}^{d} P_{2,r,\ell,\ell_2}$$

For each $i \neq j \in [n]$ and $\ell = \ell_2$, using Hoeffding inequality (see Lemma A.9), we can show

$$\Pr[|\sum_{r=1}^{m} \sum_{\ell=1}^{d} P_{1,r,\ell}| \leq 100\|F_j - y_j\|_2\|F_i - y_i\|_2 \cdot \sqrt{md\log(nd/\delta)}]$$

$$\geq 1 - \delta / \operatorname{poly}(nd).$$

Similarly, we consider $i = j$ and $\ell \neq \ell_2$ by Hanson-Wright inequality (Lemma A.10), we have

$$\Pr[|\sum_{r=1}^{m}\sum_{\ell=1}^{d}\sum_{\ell_2=1}^{d} P_{2,r,\ell,\ell_2}| \leq 100\|F_j - y_j\|_2 \|F_i - y_i\|_2 \cdot \sqrt{md}\log(nd/\delta)]$$
$$\geq 1 - \delta/\operatorname{poly}(nd).$$

By mean inequality, we have

$$\sum_{i=1}^{n}\sum_{j=1}^{n}\|F_j - y_j\|_2 \cdot \|F_i - y_i\|_2 \leq n\|F - y\|_F^2.$$

Note that by Lemma condition, we have

$$\frac{1}{m}\lambda \gtrsim \frac{n\exp(6B)}{m^2}\cdot\sqrt{md}\log(nd/\delta) \iff m \gtrsim \lambda^{-2},$$

the equation (Eq. (16) and the bound for Eq. (17)) above indicates that $\langle \operatorname{vec}(Y - F(\tau)), \operatorname{vec}(v_1)\rangle$ can be expressed as

$$\operatorname{vec}(v_1)^\top \operatorname{vec}(Y - F(\tau)) \geq 0.8m\eta \cdot \underbrace{\operatorname{vec}(F(\tau) - Y)^\top}_{1\times nd} \underbrace{H(\tau)^\top}_{nd\times nd} \operatorname{vec}(F(\tau) - Y). \qquad (18)$$

We finish the proof. $\qquad\square$

**Claim E.5.** *If the below conditions are true*

- *Let $B \geq 1$ be defined as Definition 4.1*

- *Let $\lambda = \lambda_{\min}(H^*) > 0$*

- *$C_1 = -m\eta\operatorname{vec}(F(\tau) - Y)^\top H(\tau)\operatorname{vec}(F(\tau) - Y).$*

- *$R = \lambda/(2nd\exp(10B))$*

*Then, we have*

$$C_1 \leq -\frac{1}{2}m\eta\lambda \cdot \|F(\tau) - Y\|_F^2$$

*and*

$$\lambda_{\min}(H(\tau)) \geq \lambda/2.$$

*holds with probability at least $1 - \delta$.*

*Proof.* By Lemma 5.1, with probability at least $1 - \delta$, we have

$$\|H^* - H(\tau)\|_F$$
$$\leq Rnd \cdot \exp(10B)$$
$$\leq \lambda/2 \qquad (19)$$

where the first step follows from the definition of $H(\tau)$, the last step comes from choice of $\lambda$ (see Claim Statement).

Given that $\lambda = \lambda_{\min}(H^*)$, by eigenvalue perturbation theory

$$\lambda_{\min}(H(\tau))$$
$$\geq \lambda_{\min}(H^*) - \|H^* - H(\tau)\|$$
$$\geq \lambda_{\min}(H^*) - \|H^* - H(\tau)\|_F$$
$$\geq \lambda_{\min}(H^*) - \lambda/2$$

$$\geq \lambda/2.$$

where the first step comes from triangle inequality, the second step is due to Frobenius norm, the third step is due to Eq.(19), the last step follows from $\lambda_{\min}(H^*) = \lambda$.

Finally, we have

$$\text{vec}(F(\tau) - Y)^\top H(\tau) \text{vec}(F(\tau) - Y) \geq \lambda/2 \cdot \|F(\tau) - Y\|_F^2.$$

Thus, we complete the proof. $\qquad\square$

### E.5 Bounding $C_2$

Here, we give the bound of the second order term $C_2$.

**Claim E.6.** *If the below conditions are true*

- *Let $\lambda = \lambda_{\min}(H^*)$*

- *Let $\alpha_i(\tau) := \langle \exp(W(\tau)^\top x_i), \mathbf{1}_m \rangle$*

- *Let scalar $v_{2,\ell,i} \in \mathbb{R}$ be defined as follows*

$$v_{2,\ell,i} := m \sum_{r=1}^m a_{\ell,r} \cdot \alpha_i(\tau)^{-1} \exp((\langle w_r(\tau), x_i \rangle) \cdot \eta^2 \cdot \Theta(1) \cdot \langle \Delta w_r(\tau), x_i \rangle^2$$

- *$C_2 = 2\langle \text{vec}(F(\tau) - Y), \text{vec}(v_2) \rangle$*

*Then we can conclude that*

$$C_2 \leq 2m\eta^2 n^2 d^2 \exp(9B)\|F(\tau) - Y\|_F^2.$$

*with probability at least $1 - n \cdot \exp(-mR)$.*

*Proof.* Let $p_{i,r} \in [-1, 1]$. We have

$$|v_{2,\ell,i}| = m \sum_{r\in[m]} a_{\ell,r} \cdot \mathcal{S}_{i,r} \cdot (\eta^2 p_{i,r} \langle x_i, \Delta w_r(\tau) \rangle^2)$$
$$\leq m\eta^2 nd \exp(9B)\|F(\tau) - Y\|_F^2,$$

where the last step follows Lemma D.1 and Part 11 of Lemma B.1.

Thus,

$$C_2 = 2\langle \text{vec}(F(\tau) - Y), \text{vec}(v_2) \rangle$$
$$\leq 2\|F(\tau) - Y\|_F \|v_2\|_F$$
$$\leq 2m\eta^2 n^2 d^2 \exp(9B)\|F(\tau) - Y\|_F^2,$$

where the first step follows Cauchy-Schwartz inequality, and the second step follows $\|F(\tau) - Y\|_F \leq O(\sqrt{nd})$ by induction statement (See Lemma C.3).

$\qquad\square$

### E.6 Bounding $\|F(\tau + 1) - F(\tau)\|_F^2$

Here, we give the bound of the third order term $C_3$.

**Claim E.7.** *If the below conditions are true*

- *Let $B \geq 1$ be defined as Definition 4.1*

- *$C_3 = \|F(\tau + 1) - F(\tau)\|_F^2$.*

- *$R \in (0, 0.01)$*

- **Gradient Property.** $\eta\|\Delta w_r(i)\|_2 \leq 0.01$, $\forall r \in [m]$, $\forall i \in [\tau]$

*Then with probability at least $1 - \delta$, we have*

$$C_3 \leq \eta^2 m^2 \cdot n^2 d^2 \cdot \exp(16B) \cdot \|F(\tau) - Y\|_F^2.$$

*Proof.* Note that we denote $\alpha_i$ as $\langle \mathbf{1}_m, \exp(W^\top x_i)\rangle$. According to definition of $F_{\ell,i}(\tau)$, we have

$$F_{\ell,i}(\tau + 1) - F_{\ell,i}(\tau)$$
$$= ma_\ell^\top ($$
$$\quad + \alpha_i(\tau+1)^{-1}\exp((W(\tau+1)^\top x_i) - \alpha_i(\tau)^{-1}\exp((W(\tau+1)^\top x_i)$$
$$\quad + \alpha_i(\tau)^{-1}\exp((W(\tau+1)^\top x_i) - \alpha_i(\tau)^{-1}\exp((W(\tau)^\top x_i)$$
$$)$$

Then we have

$$|F_{\ell,i}(\tau+1) - F_{\ell,i}(\tau)| \tag{20}$$
$$\leq m\sum_{r=1}^m |\alpha_i(\tau+1)^{-1} - \alpha_i(\tau)^{-1}|\exp(w_r(\tau+1)^\top x_i)$$
$$+ m\sum_{r=1}^m \alpha_i(\tau)^{-1}\exp(w_r(\tau)^\top x_i) \cdot |\exp(-\eta\Delta w_r(\tau)^\top x_i) - 1|$$

where it follows from triangle inequality.

For the second term in Eq. (20), we have

$$m\sum_{r=1}^m \alpha_i(\tau)^{-1}\exp(w_r(\tau)^\top x_i) \cdot |\exp(-\eta\Delta w_r(\tau)^\top x_i) - 1|$$
$$\leq \exp(B+R)\exp(B+R)\sum_{r=1}^m |\exp(-\eta\Delta w_r(\tau)^\top x_i) - 1|$$
$$\leq \exp(2B+2R)\sum_{r=1}^m 2\eta\|\Delta w_r(\tau)\|_2$$
$$= 2\eta\exp(2B+2R)\sum_{r=1}^m \|\Delta w_r(\tau)\|_2$$
$$\leq 2\eta\exp(2B+2R)\cdot m\cdot \exp(3B)\sqrt{nd}\|F(\tau) - Y\|_F$$
$$\leq \eta m\exp(6B)\sqrt{nd}\|F(\tau) - Y\|_F$$

where the first step comes from Lemma B.1, the second step is due to $\eta\|\Delta w_r(\tau)\|_2 \leq 0.01$ (this is stated in Claim assumption) and Fact A.8, the third step is from simple algebra, the fourth step is due to Lemma D.1, the last step follows from simple algebra.

Similarly, for the first term in Eq. (20) we have

$$m\sum_{r=1}^m |\alpha_i(\tau+1)^{-1} - \alpha_i(\tau)^{-1}|\exp(w_r(\tau+1)^\top x_i)$$
$$\leq m^2\exp(B+R)|\alpha_i(\tau+1)^{-1} - \alpha_i(\tau)^{-1}|$$
$$\leq m\exp(B+R)|\eta\Delta w_r(\tau)^\top x_i|\exp(3B+2R)$$
$$\leq \eta m\exp(4B+3R)\|\Delta w_r(\tau)\|_2$$
$$\leq \eta m\exp(7B+3R)\sqrt{nd}\|F(\tau) - Y\|_F$$

where the first step follows from Part 5 of Lemma B.1, the second step follows from Part 9 of Lemma B.1 where $R = |\eta\Delta w_r(\tau)^\top x_i|$, the third step follows from simple algebra, and the last step follows from Lemma D.1.

Thus we have

$$|F_{\ell,i}(\tau+1) - F_{\ell,i}(\tau)| \leq \eta m \exp(8B)\sqrt{nd}\|F(\tau) - Y\|_F. \tag{21}$$

Finally, we get

$$\|F(\tau+1) - F(\tau)\|_F^2 \leq nd \cdot (\eta m \exp(8B)\sqrt{nd}\|F(\tau) - Y\|_F)^2$$
$$\leq \eta^2 m^2 \cdot n^2 d^2 \cdot \exp(16B) \cdot \|F(\tau) - Y\|_F^2$$

where the first step is because of Eq. (21), the last step comes from simple algebra. □

## F  NTK REGRESSION

In this section, we introduce the NTK regression, as we will show that the neural network is "equivalent" to this regression so that we can give a final guarantee on the test data. To clarify the function, we use $F_{nn}$ to denote $F$ as a neural network function. We use $x_{te} \in \mathbb{R}^d$ to denote the test data. We would like to control the error between the neural network $F_{nn}$ and the function $F_{ntk}$. For convenience, we call this error "coupling error", which is the difference between the trained neural network and its corresponding NTK regression.

Recall that, by Definition 3.6, we have the $H^* = H(W(0))$. Recall $[H^*]_{i,j} \in \mathbb{R}^{d \times d}$ is the kernel between $x_i$ and $x_j$. Similarly, $\forall \ell_1, \ell_2 \in [d]$, for test data, we can define the NTK induced feature map as

$$[K^*_{\ell_1,\ell_2}]_{te,j} := \frac{1}{m} x_{te}^\top x_j \sum_{r=1}^m \langle v_{\ell_1,r}, \mathcal{S}_{te}(0)\rangle \cdot m\mathcal{S}_{te,r}(0) \cdot \langle v_{\ell_2,r}, \mathcal{S}_j(0)\rangle \cdot m\mathcal{S}_{j,r}(0)$$

$$[K(\tau)_{\ell_1,\ell_2}]_{te,j} := \frac{1}{m} x_{te}^\top x_j \sum_{r=1}^m \langle v_{\ell_1,r}, \mathcal{S}_{te}(\tau)\rangle \cdot m\mathcal{S}_{te,r}(\tau) \cdot \langle v_{\ell_2,r}, \mathcal{S}_j(\tau)\rangle \cdot m\mathcal{S}_{j,r}(\tau),$$

where $K^*_{te}, K_{te}(\tau) \in \mathbb{R}^{d \times nd}$. Similarly, we have $K^*_i = [H^*]_i \in \mathbb{R}^{d \times nd}, K_i(\tau) = [H(\tau)]_i \in \mathbb{R}^{d \times nd}$ for training data $x_i$. Then, we define the kernel regression predictor.

**Definition F.1** (NTK regression predictor). *We define NTK regression predictor as*

$$F_{ntk}(\gamma(\tau), x_{te}) := mK^*_{te}\gamma(\tau), \tag{22}$$

*where $\gamma(\tau) \in \mathbb{R}^{nd}$ is the parameter at timestamp $\tau$.*

Recall that we have a training dataset $\mathcal{D}_n = \{(x_i, y_i)\}_{i=1}^n$. Then, we denote the corresponding objective function for $F_{ntk}$ as

$$\mathcal{L}_{ntk}(\gamma(\tau)) = \frac{1}{2} \sum_{i=1}^n \|F_{ntk}(\gamma(\tau), x_i) - y_i\|_2^2. \tag{23}$$

Thus, based on Eq. (23), the gradient desent (GD) updating rule of $\gamma(\tau)$ is given by

$$\underbrace{\gamma(\tau+1)}_{nd \times 1} = \underbrace{\gamma(\tau)}_{nd \times 1} - \eta \cdot (m \underbrace{H^*}_{nd \times nd} \underbrace{\gamma(\tau)}_{nd \times 1} - \underbrace{\mathrm{vec}(Y)}_{nd \times 1}), \quad \gamma(0) = \mathbf{0}_{nd}, \tag{24}$$

where the Eq. (24) is according to $\gamma(\tau+1) = \gamma(\tau) - \eta\nabla_\gamma\mathcal{L}_{ntk}(\gamma(\tau))$.

### F.1  EQUIVALENCE BETWEEN TRAINED NET AND KERNEL REGRESSION

We provide a stronger bound between $F_{ntk}$ and $F_{nn}$ result compared to Lemma F.1 in (Arora et al., 2019b). Our following statement is stronger in the two following senses: their result only holds when $t \to \infty$, and our result holds for all $t \in [0, \infty)$; also their result only works for 1 dimension output space, our result holds arbitrary $d$ dimensional output space.

**Theorem F.2** (Kernel value perturbation $\Rightarrow$ prediction perturbation). *Fix $\epsilon_H \leq \frac{1}{2}\lambda$. If for all $\tau \geq 0$, $\|K_{\ell,te}^* - K_{\ell,te}(\tau)\|_F \leq \epsilon_{\ell,test}$ and $\|H^* - H(\tau)\|_F \leq \epsilon_H$, then for any $x_{te} \in \mathbb{R}^d$, $\ell \in [d]$ and $\tau \geq 0$, we have*

$$|F_{ntk}(\gamma(\tau), x_{te})_\ell - F_{nn}(W(\tau), x_{te})_\ell| \leq O\left(\frac{\sqrt{nd}}{\lambda}\epsilon_{\ell,test} + \frac{\sqrt{nd}}{\lambda^2}\log^2\left(\frac{nd}{\epsilon_H m\lambda}\right)\epsilon_H\right).$$

*Proof of Theorem F.2.* Our proof relies on a careful analysis of the trajectories induced by gradient flow for optimizing the neural network predictor $F_{nn}$ and the NTK predictor $F_{ntk}$. Then, we can have a similar argument to gradient descent at any timestamp $\tau$.

Recall that for any $x_{te}, x_i \in \mathbb{R}^d$, we have $K_{te}^*, K_i^* \in \mathbb{R}^{d \times nd}$ be the feature map induced by NTK. For any $x \in \mathbb{R}^d$, we define $\phi(x) \in \mathbb{R}^{d \times d}$ as following, for any $\ell \in [d]$,

$$\phi(x)_\ell = \frac{1}{\sqrt{m}}x\sum_{r=1}^m \langle v_{\ell,r}, \mathcal{S}(0)\rangle \cdot m\mathcal{S}_r(0).$$

We denote $\phi(X) \in \mathbb{R}^{d \times nd}$ as the stack of feature map of $X \in \mathbb{R}^{d \times n}$.

Note the optimal solution in Eq. (22) can be rewritten as

$$\min_\gamma \|\gamma\|_2 \text{ such that } mK_i^*\gamma = y_i \text{ for } i = 1, \ldots, n.$$

We have the optimal solution for kernel regression is $\gamma^* := m^{-1}(H^*)^{-1}\text{vec}(Y)$ and its corresponding prediction for $x_{te}$ will be $F_{ntk}(\gamma(\tau), x_{te}) = K_{te}^*(H^*)^{-1}\text{vec}(Y)$. The solution to this program can be rewritten as applying gradient flow on the

$$\min_\beta \sum_{i=1}^n \|\sqrt{m}\phi(x_i)^\top\beta - y_i\|_2^2$$

with initialization $\beta(0) = \mathbf{0}_d$. We use $\beta(\tau)$ to denote this parameter at timestamp $\tau$ trained by gradient flow. We denote

$$F_{ntk2}(\beta(\tau), x_{te}) := \sqrt{m}\phi(x_{te})^\top\beta(\tau)$$

where $F_{ntk2}(\beta(\tau), x_{te})$ be the predictor for $x_{te}$ at time $\tau$. Then we have

$$F_{ntk2}(\beta(\tau), x_{te}) = \sqrt{m}\underbrace{\phi(x_{te})^\top}_{\mathbb{R}^{d \times d}}\underbrace{\beta(\tau)}_{\mathbb{R}^d}$$

$$= \sqrt{m}\underbrace{\phi(x_{te})^\top}_{\mathbb{R}^{d \times d}}(\sqrt{m}\underbrace{\phi(X)}_{\mathbb{R}^{d \times nd}})\underbrace{\gamma(\tau)}_{\mathbb{R}^{nd}}$$

$$= m\underbrace{K_{te}^*}_{\mathbb{R}^{d \times nd}}\gamma(\tau)$$

$$= F_{ntk}(\gamma(\tau), x_{te})$$

where the second step follows $\beta(\tau) = \sqrt{m}\phi(X)\gamma(\tau)$ the third step follows $K_{te}^* = \phi(x_{te})^\top\phi(X)$.

With these notations, as $\tau$ goes to infinity, we denote, for any $\ell \in [d]$,

$$F_{ntk2}(x_{te})_\ell = \int_{\tau=0}^\infty \frac{\mathrm{d}F_{ntk2}(\beta(\tau), x_{te})_\ell}{\mathrm{d}\tau}\mathrm{d}\tau$$

where we have used the fact that the initial prediction is 0 as $\beta(0) = \mathbf{0}_d$. Similarly for $F_{nn}(x_{te})_\ell$. Let $F_{ntk2,i}(\tau) = F_{ntk2}(\beta(\tau), x_i)$ and $F_{ntk2}(\tau) \in \mathbb{R}^{d \times n}$. Similarly, for the NN predictor $F_{nn}$. Now we take a closer look at the time derivative:

$$\frac{\mathrm{d}F_{ntk2}(\beta(\tau), x_{te})_\ell}{\mathrm{d}\tau} = \left\langle \frac{\partial F_{ntk2}(\beta(\tau), x_{te})_\ell}{\partial\beta(\tau)}, \frac{\mathrm{d}\beta(\tau)}{\mathrm{d}\tau}\right\rangle$$

$$= \left\langle \frac{\partial F_{ntk2}(\beta(\tau), x_{te})_\ell}{\partial\beta(\tau)}, -\frac{\partial\mathcal{L}(\beta(\tau), \{x_i\}_{i=1}^n)}{\partial\beta(\tau)}\right\rangle$$

$$= -\left\langle \frac{\partial F_{ntk2}(\beta(\tau), x_{te})_\ell}{\partial \beta(\tau)}, \sum_{i=1}^{n}\sum_{\ell_2=1}^{d}(F_{ntk2,i,\ell_2}(\tau) - y_{i,\ell_2})\frac{\partial F_{ntk2}(\beta(\tau), x_i)_{\ell_2}}{\partial \beta(\tau)}\right\rangle$$

$$= -m\left\langle \phi(x_{te})_\ell, \sum_{i=1}^{n}\sum_{\ell_2=1}^{d}(F_{ntk2,i,\ell_2}(\tau) - y_{i,\ell_2})\phi(x_i)_{\ell_2}\right\rangle$$

$$= -m\,\mathrm{vec}(K^*_{\ell,te})^\top \mathrm{vec}(F_{ntk2}(\tau) - Y) \tag{25}$$

where the first step follows from simple algebra, the second step follows from ODE formulation (we remark that this is a very standard step in all the NTK literature), the third step follows from Eq. (23), the fourth step follows from the definition of $\phi(x_{te})_\ell$, the last step follows from simple algebra.

We can obtain a time derivative of the same form for $F_{nn}$.

$$\frac{\mathrm{d}F_{nn}(W(\tau), x_{te})_\ell}{\mathrm{d}\tau} = \left\langle \frac{\partial F_{nn}(W(\tau), x_{te})_\ell}{\partial W(\tau)}, \frac{dW(\tau)}{\mathrm{d}\tau}\right\rangle$$

$$= \left\langle \frac{\partial F_{nn}(W(\tau), x_{te})_\ell}{\partial W(\tau)}, -\frac{\partial \mathcal{L}(W(\tau), \{x_i\}_{i=1}^n)}{\partial W(\tau)}\right\rangle$$

$$= -\left\langle \frac{\partial F_{nn}(W(\tau), x_{te})_\ell}{\partial W(\tau)}, \sum_{i=1}^{n}\sum_{\ell_2=1}^{d}(F_{nn,i,\ell_2}(\tau) - y_{i,\ell_2})\frac{\partial F_{nn}(W(\tau), x_i)_{\ell_2}}{\partial W(\tau)}\right\rangle$$

$$= -m\,\mathrm{vec}(K_{\ell,te}(\tau))^\top \mathrm{vec}(F_{nn}(\tau) - Y) \tag{26}$$

where the first step follows from simple algebra, the second step is standard in NTK literature, the third step follows from Eq. (23), the last step follows from simple algebra.

Thus we analyze the difference between the NN predictor and NTK predictor via this integral form

$|F_{nn}(x_{te})_\ell - F_{ntk2}(x_{te})_\ell|$

$$= \left|F_{nn}(W(0), x_{te})_\ell + \int_{\tau=0}^{\infty}\left(\frac{\mathrm{d}F_{nn}(W(\tau), x_{te})_\ell}{\mathrm{d}\tau} - \frac{\mathrm{d}F_{ntk2}(\beta(\tau), x_{te})_\ell}{\mathrm{d}\tau}\right)\mathrm{d}\tau\right|$$

$$= \left|F_{nn}(W(0), x_{te})_\ell| + \left|-m\int_{\tau=0}^{\infty}\left(\mathrm{vec}(K_{\ell,te}(\tau))^\top \mathrm{vec}(F_{nn}(\tau) - Y) - \mathrm{vec}(K^*_{\ell,te})^\top \mathrm{vec}(F_{ntk2}(\tau) - Y)\right)\mathrm{d}\tau\right|$$

$$= \left|-m\int_{\tau=0}^{\infty}\left(\mathrm{vec}(K_{\ell,te}(\tau))^\top \mathrm{vec}(F_{nn}(\tau) - Y) - \mathrm{vec}(K^*_{\ell,te})^\top \mathrm{vec}(F_{ntk2}(\tau) - Y)\right)\mathrm{d}\tau\right|$$

$$\leq m\left|\int_{\tau=0}^{\infty}\mathrm{vec}(K_{\ell,te}(\tau) - K^*_{\ell,te})^\top \mathrm{vec}(F_{nn}(\tau) - Y)\mathrm{d}\tau\right| + m\left|\int_{\tau=0}^{\infty}\mathrm{vec}(K^*_{\ell,te})^\top \mathrm{vec}(F_{nn}(\tau) - F_{ntk2}(\tau))\mathrm{d}\tau\right|$$

$$\leq m\max_{0\leq t\leq\infty}\|K_{\ell,te}(\tau) - K^*_{\ell,te}\|_F\int_{\tau=0}^{\infty}\|F_{nn}(\tau) - Y\|_F\mathrm{d}\tau + m\max_{0\leq t\leq\infty}\|K^*_{\ell,te}\|_F\int_{\tau=0}^{\infty}\|F_{nn}(\tau) - F_{ntk2}(\tau)\|_F\mathrm{d}\tau$$

$$\leq m\epsilon_{\ell,test}\int_{\tau=0}^{\infty}\|F_{nn}(\tau) - Y\|_F\mathrm{d}\tau + m\max_{0\leq t\leq\infty}\|K^*_{\ell,te}\|_F\int_{\tau=0}^{\infty}\|F_{nn}(\tau) - F_{ntk2}(\tau)\|_F\mathrm{d}\tau,$$

where the first step follows from the difference between the NN predictor and NTK predictor, the second step follows from Eq. (25) and Eq. (26), the third step follows $|F_{nn}(W(0), x_{te})_\ell| = 0$ by symmetric initialization from Definition 3.7, the fourth step follows from simple algebra, the fifth step follows from Frobenius norm, the last step follows from simple algebra.

For the first term, recall $\|H^* - H(\tau)\|_F \leq \epsilon_H$ and, by Claim E.5, we have

$$\lambda_{min}(H(\tau)) \geq \frac{1}{2}\lambda.$$

Using this fact we know $\|F_{nn}(\tau) - Y\|_F \leq \exp(-\frac{m}{2}\lambda\tau)\|F_{nn}(0) - Y\|_F$ (The reason to obtain this is due to solve ODE).

Therefore, by Lemma D.3, we can bound

$$\int_{\tau=0}^{\infty}\|F_{nn}(\tau) - Y\|_F\mathrm{d}\tau = \int_{\tau=0}^{\infty}\exp\left(-\frac{m}{2}\lambda\tau\right)\|F_{nn}(0) - Y\|_F\mathrm{d}\tau$$

$$= O(\frac{\sqrt{nd}}{m\lambda}).$$

To bound $\int_{\tau=0}^{\infty} \|F_{nn}(\tau) - F_{ntk2}(\tau)\|_F d\tau$, we observe that $F_{nn}(\tau) \to y$ and $F_{ntk2}(\tau) \to y$ with linear convergence rate. Therefore, we can choose some $\tau_0 = \frac{C}{m\lambda} \log\left(\frac{nd}{\epsilon_H \cdot m\lambda}\right)$ so that

$$\int_{\tau_0}^{\infty} \|F_{nn}(\tau) - F_{ntk2}(\tau)\|_F d\tau \leq \int_{\tau_0}^{\infty} \|F_{nn}(\tau) - Y\|_F d\tau + \int_{\tau_0}^{\infty} \|F_{ntk2}(\tau) - Y\|_F d\tau$$

$$\leq O\left(\frac{1}{m\lambda}(\|F_{nn}(\tau_0) - Y\|_F + \|F_{ntk2}(\tau_0) - Y\|_F)\right)$$

$$\leq O\left(\frac{\sqrt{nd}}{m\lambda} \exp\left(-m\lambda\tau_0\right)\right)$$

$$\leq O(\epsilon_H).$$

where the first step follows from simple algebra, the second step follows from integral range is $\tau_0$, the third step follows from Lemma D.3, the last step follows from choice of $\tau_0$.

Thus it suffices to bound $\int_{\tau=0}^{\tau_0} \|F_{nn}(\tau) - F_{ntk2}(\tau)\|_F d\tau \leq \tau_0 \max_{0 \leq t \leq \tau_0} \|F_{nn}(\tau) - F_{ntk2}(\tau)\|_F$.

First observe that

$$\|F_{nn}(\tau) - F_{ntk2}(\tau)\|_F \leq \|F_{nn}(0)\|_F + \int_{s=0}^{\tau} \left\|\frac{d(F_{nn}(s) - F_{ntk2}(s))}{ds}\right\|_F ds$$

$$= \int_{s=0}^{\tau} \left\|\frac{d(F_{nn}(s) - F_{ntk2}(s))}{ds}\right\|_F ds,$$

where the last step follows symmetric initialization from Definition 3.7.

Note

$$\frac{d(F_{nn}(\tau) - F_{ntk2}(\tau))}{d\tau} = -mH(\tau) \text{vec}(F_{nn}(\tau) - Y) + mH^* \text{vec}(F_{ntk2}(\tau) - Y)$$

$$= -mH^* \text{vec}(F_{nn}(\tau) - F_{ntk2}(\tau)) + m(H^* - H(\tau)) \text{vec}(F_{nn}(\tau) - Y)$$

where the first step follows from definition of $F_{nn}$ and $F_{ntk2}$.

Since $H^*$ is positive semidefinite, $-H^* \text{vec}(F_{nn}(\tau) - F_{ntk2}(\tau))$ term only makes $\|F_{nn}(\tau) - F_{ntk2}(\tau)\|_F$ smaller. Therefore, we have

$$\|F_{nn}(\tau) - F_{ntk2}(\tau)\|_F \leq m \int_{s=0}^{\tau} \|F_{nn}(s) - Y\|_F \|H(\tau) - H^*\|_F ds$$

$$\leq m\tau \|F_{nn}(0) - Y\|_F \epsilon_H$$

$$\leq O\left(m\tau\sqrt{nd}\epsilon_H\right),$$

where the last step is by Lemma D.3.

Therefore, we have

$$\int_{\tau=0}^{\tau_0} \|F_{nn}(\tau) - F_{ntk2}(\tau)\|_F d\tau \leq O\left(m\tau_0^2 \sqrt{nd}\epsilon_H\right)$$

$$= O\left(\frac{\sqrt{nd}}{m\lambda^2} \log^2\left(\frac{nd}{\epsilon_H m\lambda}\right) \epsilon_H\right).$$

where the first step follows from integral range is $\tau_0$, the second step follows from the choice of $\tau_0$.

Lastly, as $F_{ntk2}(x_{te})_\ell = F_{ntk}(x_{te})_\ell$, we put things together and get

$$|F_{ntk}(x_{te})_\ell - F_{nn}(x_{te})_\ell| \leq O\left(\frac{\sqrt{nd}}{\lambda}\epsilon_{\ell,test} + \frac{\sqrt{nd}}{\lambda^2} \log^2\left(\frac{nd}{\epsilon_H m\lambda}\right) \epsilon_H\right).$$

From the above, after we change the integration from $(0, \infty)$ to $(0, \tau)$, the statement still holds. Then, based on the gradient flow version, we can have a gradient descent version with a constant error factor by replacing integral with geometric summarization (for example $\sum_{i=0}^{\infty} a^i < 2$, when $a \in (0, 0.5)$ ). $\qquad\square$

## G  DIFFUSION

In Section G.1, we provide the proof of our main result of diffusion. In Section G.2, we provide some tools from previous works.

We first define an auxiliary function $\widetilde{F}_{ntk}$ of the same functional form as $F_{ntk}$, but trained on a pseudo dataset $\widetilde{S} := \{\widetilde{y}_i, x_i\}_{i=1}^n$ with $\widetilde{y}_i := F_{\mathcal{H}}(x_i) + \epsilon_i$ and $\epsilon_i := y_i - F_*(x_i)$. Then, we have the following claim.

**Claim G.1** (Loss decomposition). *We can decompose our target function as the following*

$$\frac{1}{T} \int_0^T \mathbb{E}[\|F_{nn}(W(\tau), (t, x(t))) - F_*(t, x(t))\|_2^2]\mathrm{d}t \leq Z_1 + Z_2 + Z_3 + Z_4,$$

*where*

$$Z_1 = \frac{1}{T} \int_0^T \mathbb{E}[\|F_{nn}(W(\tau), (t, x(t))) - F_{ntk}(\gamma(\tau), (t, x(t)))\|_2^2]\mathrm{d}t \qquad \textit{(coupling)}$$

$$Z_2 = \frac{1}{T} \int_0^T \mathbb{E}[\|F_{ntk}(\gamma(\tau), (t, x(t))) - \widetilde{F}_{ntk}(\gamma(\tau), (t, x(t)))\|_2^2]\mathrm{d}t \qquad \textit{(label mismatch)}$$

$$Z_3 = \frac{1}{T} \int_0^T \mathbb{E}[\|\widetilde{F}_{ntk}(\gamma(\tau), (t, x(t))) - F_{\mathcal{H}}(t, x(t))\|_2^2]\mathrm{d}t \qquad \textit{(early stopping)}$$

$$Z_4 = \frac{1}{T} \int_0^T \mathbb{E}[\|F_{\mathcal{H}}(t, x(t)) - F_*(t, x(t))\|_2^2]\mathrm{d}t. \qquad \textit{(approximation)}.$$

The coupling error term is the gap between neural networks $F_{nn}$ and a kernel function $F_{ntk}$. The approximation error term is the gap between the target function $F_*$ and its corresponding RKHS function $F_H$. These two terms transfer the problem of neural networks training into the problem of kernel regression.

### G.1  MAIN RESULT OF DIFFUSION

In this section, we prove the main result of diffusion.

**Theorem G.2** (Restatement of Theorem 6.6). *Suppose Assumptions 6.1, 6.2, 6.3, 6.4 hold and we set $m = \Omega(\lambda^{-2}n^3d^3\exp(18B)\log^2(nd/\delta))$ and $\eta = 0.1\lambda/(mn^2d^2\exp(16B))$. Moreover, suppose $\widehat{T}$ satisfies Assumption 6.5 with corresponding $\epsilon(n, \widehat{T})$. Then for large enough $R_{\mathcal{H}}$, with probability at least $1 - \delta$, it holds that*

$$\frac{1}{T} \int_0^T \lambda(t)\mathbb{E}[\|s_{W(\widehat{T})}(t, x(t)) - \nabla \log p_t(X_t)\|_2^2]\mathrm{d}t$$

$$\leq O\left(\frac{1}{\lambda\sqrt{n}} + \epsilon(n, \widehat{T}) + dA^2(R_{\mathcal{H}}) + dA(R_{\mathcal{H}}) + \sqrt{dA(R_{\mathcal{H}})\Gamma_\delta} + \Gamma_\delta\right).$$

*Proof of Theorem 6.6.* Note that the $m$ and $\eta$ satisfy the conditions in Theorem 4.2. The reason about a different $m$ is that we choose a different $R$ and apply Lemma E.2 one more time. Recall the $\epsilon_{\ell,test}$ and $\epsilon_H$ are defined in Theorem F.2.

Note that $H^* = H(0)$. By Lemma 5.1, Part 2, let $R = \lambda/(2n^2d^2\exp(10B))$, we have with probability at least $1 - \delta$ such that

$$\| \underbrace{H^*}_{nd \times nd} - \underbrace{H(\tau)}_{nd \times nd} \|_F \leq \epsilon_H = \frac{\lambda}{2nd}.$$

Note that $K_{\ell,te}^*$ and $K_{\ell,te}$ share the same weight perturbation as $H^*$ and $H(\tau)$. Thus, by using the same proof as Lemma 5.1, Part 1, we have

$$\| \underbrace{K_{\ell,te}^*}_{n \times d} - \underbrace{K_{\ell,te}}_{n \times d} \|_F \leq \epsilon_{\ell,test} = \frac{\lambda}{2n^{1.5}d^{1.5}}.$$

We have

$$\|F_{ntk}(\gamma(\tau), x_{te}) - F_{nn}(W(\tau), x_{te})\|_2$$
$$\leq \sqrt{d} \max_{\ell \in d} |F_{ntk}(\gamma(\tau)_\ell, x_{te}) - F_{nn}(W(\tau), x_{te})_\ell|$$
$$\leq O\left( \frac{\sqrt{n}d}{\lambda} \max_{\ell \in [d]} \epsilon_{\ell,test} + \frac{\sqrt{n}d}{\lambda^2} \log^2\left( \frac{nd}{\epsilon_H m\lambda} \right) \epsilon_H \right)$$
$$\leq O\left( \frac{\sqrt{n}d}{\lambda} \frac{\lambda}{n^{1.5}d^{1.5}} + \frac{\sqrt{n}d}{\lambda^2} \log^2\left( \frac{nd}{m\lambda} \right) \frac{\lambda}{nd} \right)$$
$$\leq O\left( \frac{1}{\lambda\sqrt{n}} \log^2\left( \frac{nd}{m\lambda} \right) \right)$$
$$\leq O\left( \frac{1}{\lambda\sqrt{n}} \right)$$

where the first step follows from simple algebra, the second step is by Theorem F.2.

Thus, we finish the proof by Claim G.1, where coupling is from above, label mismatch is from Theorem G.4, early stopping is from Assumption 6.5 and approximation is from Theorem G.3. $\quad\square$

## G.2 TOOLS FROM PREVIOUS WORKS

We have the following statements from previous works (Han et al., 2024b).

**Theorem G.3** (Theorem 3.6 in (Han et al., 2024b), universal approximation of score function). *Suppose Assumptions 6.1, 6.3 and 6.4 hold. Let $R_\mathcal{H}$ be larger than a constant $c_1$, i.e., $C(d+1, 0)$ in Proposition 6 of (Bach, 2017), which depends only on $d$. There exists a function $F_\mathcal{H} \in \mathcal{H}$ such that $\|F_\mathcal{H}\|_\mathcal{H}^2 \leq dR_\mathcal{H}$ and*

$$\frac{1}{T} \int_0^T \mathbb{E}[\|F_\mathcal{H}(t, x(t)) - F_*(t, x(t))\|_2^2] \mathrm{d}t \leq dA^2(R_\mathcal{H}).$$

**Theorem G.4** (Theorem 3.10 in (Han et al., 2024b), label mismatch). *Suppose Assumptions 6.1 and 6.2 hold. If we initialize both $F_{ntk}$ and $\widetilde{F}_{ntk}$ properly, then with probability at least $1 - \delta$ it holds simultaneously for all $\tau$ that*

$$\frac{1}{T} \int_0^T \mathbb{E}[\|F_{ntk}(\gamma(\tau), (t, x(t))) - \widetilde{F}_{ntk}(\gamma(\tau), (t, x(t)))\|_2^2] \mathrm{d}t$$
$$\leq dA(R_\mathcal{H}) + C_0(\sqrt{dA(R_\mathcal{H})\Gamma_\delta} + \Gamma_\delta)$$

*where $C_0$ is a constant defined in Theorem 1 of (Reeve & Kaban, 2020).*

## H DISCUSSION

In this section, we provide discussions about the potential extensions of our method on various popular frameworks, such as attention mechanism (Section H.1) and feature learning (Section H.2).

### H.1 SELF-ATTENTION LEARNING

The self-attention can be written as

$$F(W^K X, W^Q X, W^V X) \in \mathbb{R}^{d \times n'}, \tag{27}$$

where $W^K, W^Q, W^V \in \mathbb{R}^{d \times d}$ denotes key, query, and value matrix respectively and $X \in \mathbb{R}^{d \times n'}$ is a sequence of $n'$ tokens. As our work is a first step to understanding softmax, it is natural to consider how to extend our results to self-attention. It is well-known that using two reformulation tricks: tensor-trick and SVM-trick (Gao et al., 2023b;c; Alman & Song, 2024a), any analysis for softmax function can be naturally generalized to attention function $F(W^K X, W^Q X, W^V X)$. Therefore, we conjecture that we can borrow the idea from (Gao et al., 2023b;c; Alman & Song, 2024a) to decouple Eq (27) into the value term and the softmax term. And, we can alternatively optimize the weights for the softmax term $(W^k, W^Q)$ and the value term $(W^V)$. We leave this valuable direction as a future work.

## H.2    Feature Learning

Recently, there is a line of work showing that feature learning may be beyond NTK on sample complexity or time complexity, e.g., (Allen-Zhu & Li, 2019; Wei et al., 2019; Hanin & Nica, 2019; Allen-Zhu et al., 2019a; Daniely & Malach, 2020; Chen et al., 2020; Yang & Hu, 2020; Huang & Yau, 2020; Li et al., 2020; Ghorbani et al., 2020; Refinetti et al., 2021; Malach et al., 2021; Luo et al., 2021; Damian et al., 2022; Shi et al., 2022; 2024) and many more. It is worth studying the feature learning ability of two-layer softmax NN to figure out what feature pattern the softmax prefers to learn and how it happens. We leave this valuable direction as a future work.

## LLM Usage Disclosure

LLMs were used only to polish language, such as grammar and wording. These models did not contribute to idea creation or writing, and the authors take full responsibility for this paper's content.

