# OpenReview forum: "Exploring the Frontiers of Softmax: Provable Optimization, Applications in Diffusion Model, and Beyond"
_ICLR.cc/2026/Conference — Submitted to ICLR 2026_

### Official Review · Reviewer_dU7o · 2025-10-24

**Soundness:** 3
**Presentation:** 2
**Contribution:** 3
**Rating:** 4
**Confidence:** 1

**Summary:**

This paper provides a theoretical study of softmax neural networks, aiming to explain what makes softmax-based architectures successful. Using the Neural Tangent Kernel (NTK) framework, the authors analyze a two-layer softmax neural network and prove convergence and generalization guarantees in the overparameterized regime. The paper argues that the normalization property of softmax induces  a large convex region in the loss landscape. To showcase applicability, the analysis is extended to diffusion models.

**Strengths:**

1. The paper takes on an important theoretical question: understanding why softmax works so well in modern architectures.
connected to previous NTK results for ReLU and exponential activations.

2. The extension to score-based diffusion models demonstrates practical relevance and bridges theoretical and generative modeling literature.

**Weaknesses:**

1. The paper is mathematically dense but conceptually underexplained. The intuition for why softmax’s normalization leads to better generalization could be emphasized more. The paper is hard to read.

2. The “application” section is purely analytical — no experiments, no empirical validation. While the theoretical link is elegant, it remains disconnected from practice. For ICLR, empirical support or simulations demonstrating the claimed advantage would be expected.

**Questions:**

1. How does the normalization-induced “convex region” in softmax NTK differ quantitatively from the ReLU NTK case?

2. Can you empirically validate the theoretical claims on small synthetic datasets (e.g., compare softmax vs. ReLU NTK behavior)?

---

### Official Review · Reviewer_rQ1Y · 2025-10-27

**Soundness:** 3
**Presentation:** 2
**Contribution:** 2
**Rating:** 4
**Confidence:** 2

**Summary:**

This work studies property of two-layer softmax NN. Specifically, the authors exploit NTK tool to analyze convergence of two-layer softmax NN, and extends the result to diffusion model as an application.

**Strengths:**

The paper studies properties of two-layer softmax NN. The topic is worth exploring since softmax is playing important role in modern AI systems.

The authors have established convergence result in Theorem 4.2, based on NTK theoretic tools.

The authors have also extended their theoretic results to diffusion models, and establish convergence there.

**Weaknesses:**

It seems that the main contribution of the current paper is establishing convergence results for two-layer softmax NN and extends the result to diffusion model. The analysis is standard NTK analysis (though in Section 5.1the authors explicitly explain what are the new challenges in their theoretic derivation).

Also I personally feel the paper can largely benefit from adding more explanation, discussion, and comparisons. The current presentation is too notation-heavy, and lacks explanation of intuition and general understanding beyond just convergence. Some of highlight points are not well-justified. For example, in the abstract, the authors state "normalization effect of the softmax function leads to a good perturbation property of the induced NTK matrix, resulting in a good convex region of the loss landscape". But in main text, there is little verbal explanation beyond notation-heavy theorems. I feel the current presentation can improve a lot if the authors add more explanations about high-level intuition, compare to focusing more on technical details.

Some notation confusions exist.

**Questions:**

In Table 1 and and Lemma 5.1, there is comparison with ReLU and exp activation. The result seems that all three activations: ReLU, exp, and softmax, have same convergence and perturbation properties. Can I interpret it as all three  activations have similar performance from NTK perspective? Then if we replace softmax, by ReLU or exp, we will still get similar results? Also, the authors mention the key to softmax is "the normalization effect of the denominator", while ReLU and exp don't have this same denominator, their effectiveness are  drawn by some other mechanisms. A more comprehensive comparison between the underline mechanisms will be interesting.

minor errors?:

line 1033-1035: there are questions marks not rendered correctly

line 83-84: people usually write token embedding dimension as $n'\times d$ instead of $d\times n'$, which is a bit hard to follow.

line 199-203: what is $d$ here? is it $d_1=d_2=d$?

line 204: it should be $x_i\in\mathbb R^d, y_i\in\mathbb R^d$ instead of $x\in\mathbb R^d, y\in\mathbb R^d$?

line 237: a close bracket is missing in $d\mathcal L(W(\tau))$

not sure why Remark 5.2 is involved...

---

### Official Review · Reviewer_YdFW · 2025-11-03

**Soundness:** 2
**Presentation:** 1
**Contribution:** 2
**Rating:** 2
**Confidence:** 4

**Summary:**

- This paper studies the property of the softmax activation function in the two-layer network setting.
- Especially, the authors utilize the neural tangent kernel framework to theoretically analyze the perturbation property.
- To demonstrate their theoretical analysis in the practical settings, the authors apply their work in the diffusion models.

**Strengths:**

- The paper is theoretically grounded.

**Weaknesses:**

- The paper is unnecessarily over-complicated.
- In fact, LLMs can be the use case of the proposed work, but there seems to be no need to give a long explanation of LLMs. I recommend removing Sections 2.2 and 2.3 as well. Instead, the authors can explain some works, such as  Munteanu et al. (2022), in the related work section, since it is more directly linked to the proposed work.
- In Table 1, I disagree with Line 57-59: We can see that ... For example, $n^2$ and $n^{2+o(1)}$ can be hugely different.
- Some synthetic experiments should be conducted to verify how the theoretical analysis connects to the practical setting.

**Questions:**

- Do assumptions in Section 6 always hold for the diffusion models?
- The authors have assumed a two-layer network, but the noise prediction score network in the diffusion models is very complicated, with U-Net architectures. How does this gap reduce in the (proof of) Theorem 6.6?
- What can be claimed in the LLMs, similarly to Theorem 6.6?

---

### Official Review · Reviewer_uDsP · 2025-11-03

**Soundness:** 3
**Presentation:** 2
**Contribution:** 2
**Rating:** 6
**Confidence:** 4

**Summary:**

This paper studies the convergence of GD-trained transformer with softmax activate under the NTK framework. As an application, the score matching problem in diffusion models.

**Strengths:**

1. Under the NTK framework, the authors establish the convergence rate for training softmax transformers using gradient descent.
2. By leveraging the connection with score matching in diffusion models and multi-label regression, the authors obtain the convergence rate of training score functions.

**Weaknesses:**

While technical ideas make sense to me, I still have the following concerns:

1. Some literature shows that softmax transformer has better sequence length dependence compared with ReLU networks. Do you results also support this point? \
2. Any technical difficulties when applying the NTK techniques for softmax activation compared to ReLU networks? This is not clearly demonstrated in the manuscript.

**Questions:**

I have minor questions and suggestions:

1. The connection between score matching and regression is not clear. Although you cited the literature Han et al. 2024, I would appreciate it if you could make the writing even more clear.
2. Assumption 6.1, I understand $ y_i $ can be bounded as $ y_i = \mathbb{E}[x_0 \vert x_t] $ and $ x_0 $ can be bounded. However, $ x_i $ is the noisy latent obtained by adding noise. Why could we also assume that is bounded?

---

### Meta-Review · Area_Chair_Y1tY · 2026-01-06

**Summary:**

The authors study the training loss landscape of a two layer softmax network, where the authors showed there is a convex region of the landscape. The authors intend the toy model network to be a method of studying the softmax mechanism in self attention networks.

Reviewers broadly speaking were skeptical of this work, and has raised the following concerns:
- The analysis were relying on standard NTK techniques, and drawing similar conclusions re: convexity
- Notation was dense and presentation is not clear
- Convex region claim lacking comparison with the standard non-normalized results
- Weak connection to practice (LLMs and diffusion)

Overall, I tend to agree with the reviewers, although I'm not particularly bothered by the mathematical notation. The main issues in my opinion is that the model is too much of a toy model for two reasons:
1. The network architecture is really far from attention, and even further from diffusion models
2. NTK analysis typically fails to draw interesting conclusions about feature learning

In particular, I tend to agree that the analysis provided in this submission is not sufficient significantly of an improvement over the typical NTK analysis, which also yields convexity results. Furthermore, it is well understood by now the limitations of this approach, and how little we can learn about the actual network architecture.

That being said, I believe the philosophy of seeking a toy model that can capture **important properties** of attention and diffusion can be a fruitful direction of research. However, one should be realistic about the model in that it **necessarily** needs to capture some important properties that are of interest. For example, we should be able to see how this toy model behaves like attention/diffusion in a significant way, or it's a directly studying the attention mechanism on a simpler task. Furthermore, we should also be able to capture **feature learning** in this toy model, as linearization of the network really renders the interesting aspect of deep learning irrelevant.

For these reasons, I would recommend reject for this submission.

**Reviewer Concerns:**

The authors did not respond to the reviews.

**Reviewer Scores:**

uDsP: 6
YdFW: 2
rQ1Y: 4
dU7o: 4

I believe the scores would have stayed the same given the authors did not respond.

---

### Decision · Program_Chairs · 2026-01-26

Reject